# Nociceptor neurons control pollution-mediated neutrophilic asthma

Jo-Chiao Wang[1], Amelia Kulle[2], Theo Crosson[1], Amin Reza Nikpoor[3], Surbhi Gupta[3], Anais Roger[1,3], Moutih Rafei[1], Ajitha Thanabalasuriar[2,4], Sebastien Talbot[3,5]*

[1]Department of Pharmacology and Physiology, University de Montreal, Montreal, Canada; [2]Department of Microbiology and Immunology, McGill University, Montreal, Canada; [3]Department of Biomedical and Molecular Sciences, Queen's University, Kingston, Canada; [4]Department of Pharmacology and Therapeutics, McGill University, Montreal, Canada; [5]Department of Physiology and Pharmacology, Karolinska Institutet, Stockholm, Sweden

## eLife Assessment

This **important** work shows that fine particulate matter exposure to the lungs led to nociceptor-dependent neutrophilic inflammation. Likely macrophage-neuronal crosstalk, via release of artemin from macrophages and activation of Gfra3 on the JNC neuron, potentiated the response. The data **convincingly** strengthens links between pollutants, immune and neural interactions.

**\*For correspondence:**
sebastien.talbot@ki.se

**Abstract** The immune and sensory nervous systems, having evolved in parallel, communicate through shared receptors and transmitters to maintain homeostasis and respond to both external and internal disruptions. Although neural responses often confer protective benefits, they can also exacerbate inflammation during allergic reactions such as asthma. In our study, we modeled pollution-exacerbated asthma by exposing mice to ambient $PM_{2.5}$ particles alongside ovalbumin. Compared to exposure to ovalbumin alone, this co-exposure significantly increased the numbers of neutrophils and γδ T cells in bronchoalveolar lavage fluid and lung tissue, respectively. We found that silencing nociceptor neurons at the peak of inflammation using intranasal QX-314 or ablating *Trpv1*-expressing neurons reduced lung neutrophil accumulation. Live in vivo intravital imaging confirmed that neuronal ablation reduced neutrophil numbers and increased their net displacement capacity. In neurons isolated from mice with pollution-exacerbated asthma, the chemical-sensing TRPA1 channel exhibited heightened sensitivity to its cognate ligand. Elevated levels of artemin were detected in the bronchoalveolar lavage fluid of pollution-exposed mice but returned to baseline in mice with ablated nociceptor neurons. Alveolar macrophages expressing the pollution-sensing aryl hydrocarbon receptor were identified as a putative source of artemin following exposure to $PM_{2.5}$. This molecule enhanced TRPA1 responsiveness and, in turn, drove nociceptor-mediated neutrophil recruitment, revealing a novel mechanism by which lung-innervating neurons respond to air pollution in the context of allergy. Overall, our findings suggest that targeting artemin-driven pathways could provide a therapeutic strategy for controlling neutrophilic airway inflammation in asthma, a clinical condition typically refractory to treatment.

## Introduction

Organisms have evolved sophisticated fail-safe systems to preserve homeostasis, integrating threat detection, reflex responses, and tailored immune mechanisms (*Talbot et al., 2016*). These systems

are jointly orchestrated by the immune and sensory nervous systems, which are designed to sense external threats and internal disturbances. Both systems rely on shared metabolic pathways and signaling molecules—including receptors, cytokines, and neuropeptides (*Foster et al., 2015*). This common molecular framework supports constant interactions between these systems, which play pivotal roles in monitoring barrier tissues (e.g. skin [*Michoud et al., 2021*], lung [*Baral et al., 2018*; *Yang et al., 2024*], and gut [*Lai et al., 2020*]), activating anticipatory immune responses (*Cohen et al., 2019*) and managing pathologies such as allergies (*Perner et al., 2020*; *Hanč et al., 2023*), infections (*Chiu et al., 2013*), and malignancies (*Balood et al., 2022*; *Restaino et al., 2023*; *Barr et al., 2024*; *Amit et al., 2024*).

In the context of lung disease, vasoactive intestinal peptide (VIP), released by pulmonary sensory neurons following IL-5 stimulation, fosters allergic inflammation by acting on CD4$^+$ T cells and innate lymphoid type 2 cells (ILC2s). This action elevates T helper type 2 (T$_h$2) cytokines, which are central drivers of asthmatic responses (*Foster et al., 2017*; *Talbot et al., 2015*). Expanding on these insights, our work shows that vagal nociceptor neurons contribute to airway hyperresponsiveness (*Talbot et al., 2016*; *Tränkner et al., 2014*) mucus metaplasia (*Talbot et al., 2020*; *Yang et al., 2022*), and the detection of immunoglobulins E and G (*Crosson et al., 2021*; *Bersellini Farinotti et al., 2019*). These neurons also mediate antibody class switching in B cells (*Mathur et al., 2021*) and promote IgG production (*Tynan et al., 2024*). More recently, we discovered that a subset of nociceptors is reprogrammed by the asthma-associated cytokine IL-13, rendering them susceptible to the inhibitory influence of neuropeptide Y (NPY) secreted by lung-innervating sympathetic neurons (*Crosson et al., 2024*).

Wildfire frequency and severity increased by 2.2-fold within the last 20 years (*Cunningham et al., 2024*). Between 2008 and 2018 in California, wildfire-attributable PM$_{2.5}$ exposure caused an estimated ~53,000 premature deaths (*Connolly et al., 2024*). Combined with urban pollution, these events have driven escalating levels of particulate matter (PM$_{2.5}$), shifting asthma phenotypes from more responsive T$_h$2/eosinophilic types to treatment-resistant neutrophilic and mixed T$_h$17/T$_h$1 variants (*Ray and Kolls, 2017*; *Baan et al., 2022*; *de Nijs et al., 2013*; *Gianniou et al., 2018*; *Noah et al., 2023*; *Wilgus and Merchant, 2024*). Thus, recent estimates suggest that neutrophilic asthma accounts for 15–25% of all asthma cases, with about half of these resistant to standard therapies (*Jatakanon et al., 1999*).

To investigate whether neuronal silencing can mitigate neutrophilic airway inflammation, we used an innovative asthma model that combines ovalbumin (OVA) with fine particulate matter (FPM). We further explored the molecular mechanisms underlying heightened neuronal sensitivity in this setting, identifying artemin—produced by alveolar macrophages (AMs) via the pollution-sensing aryl hydrocarbon receptor (AhR)—as a critical mediator of this hypersensitivity.

## Results

### Heightened nociceptor sensitivity in pollution-exacerbated asthma

Prof. Ya-Jen Chang's group developed a novel method to model pollution-exacerbated asthma and investigate the impact of FPM on airway inflammation (*Thio et al., 2022*). Their work demonstrates that FPM exposure induces airway hyperreactivity and neutrophilic inflammation, promotes T$_h$1 and T$_h$17 immune responses, and increases epithelial cell apoptosis rates (*Thio et al., 2022*). Notably, γδ T cells significantly contribute to both the inflammation and airway hyperreactivity phenotypes by producing IL-17A (*Thio et al., 2022*). Building on this foundation, we explored whether lung-innervating nociceptor neurons become sensitized during such asthma exacerbations. To test this, male and female C57BL/6 mice (6–10 weeks of age) were sensitized with an intraperitoneal (i.p.) injection of OVA (200 µg/dose) and aluminum hydroxide (1 mg/dose) emulsion on days 0 and 7, followed by intranasal challenges of OVA (50 µg/dose), with or without FPM (20 µg/dose), on days 14–16. On day 17, lung-innervating jugular-nodose-complex (JNC) neurons were harvested, cultured for 24 hr, and loaded with the calcium indicator Fura-2AM (*Figure 1A*). Stimulation with the TRPA1 agonist AITC (10–100 µM) and the pan-neuronal activator KCl (40 mM) revealed increased responsiveness in neurons from mice exposed to both FPM and OVA, compared to OVA-only-exposed mice, whose sensitivity was similar to that of vehicle (phosphate-buffered saline [PBS]) controls (*Figure 1B and C*).

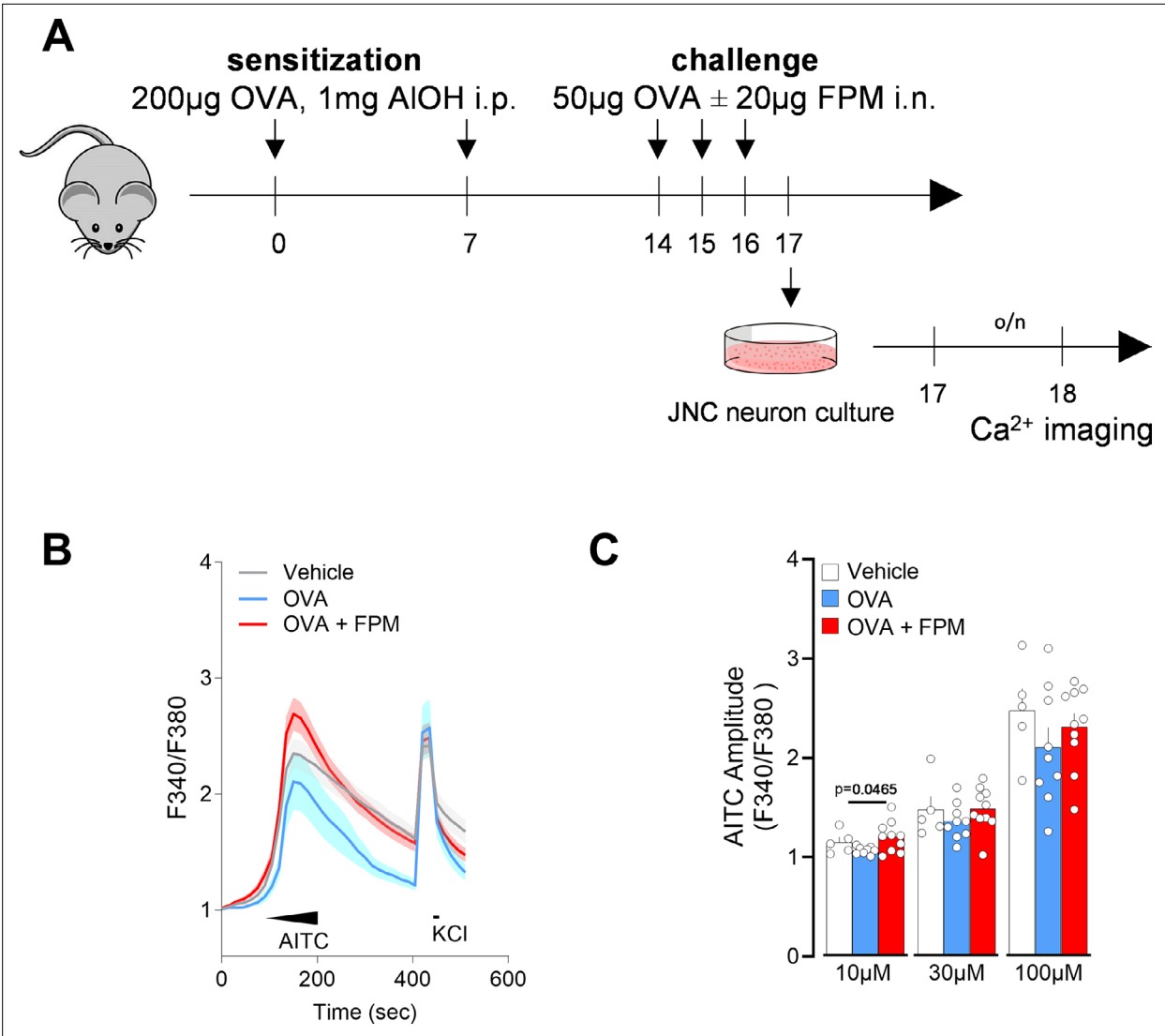

**Figure 1.** Air pollution exacerbates nociceptor neuronal activity. (**A–C**) Male and female C57BL/6 mice (6–10 weeks of age) were sensitized via intraperitoneal injection with an emulsion of ovalbumin (OVA; 200 µg/dose) and aluminum hydroxide (1 mg/dose) on days 0 and 7. On days 14–16, mice were challenged intranasally with OVA (50 µg/dose), either alone or in combination with fine particulate matter (FPM; 20 µg/dose). Bronchoalveolar lavage fluid was collected, and jugular-nodose complex neurons were cultured on day 17 for 24 hr before being loaded with the calcium indicator Fura-2AM. Cells were sequentially stimulated with the TRPA1 agonist AITC (successively to 10 µM at 60–90 s, 30 µM at 90–120 s, 100 µM at 120–150 s) and then with KCl (40 mM at 420–435 s). Calcium flux was continuously monitored throughout the experiment. The amplitude of AITC responses was measured by calculating the ratio of peak F340/F380 fluorescence after stimulation to the baseline F340/F380 fluorescence measured 30 s prior to stimulation. Data are plots as the per dish average of AITC and KCl responsive neurons and show that AITC (10 µM) responses were higher in JNC neurons from OVA-FPM-exposed mice when compared to vehicle or OVA alone (**C**). Data in are presented as means ± SEM (**B–C**). N are as follows: (**B**) n=35 neurons (control group), 19 neurons (OVA group), and 38 neurons (OVA + FPM group), (**C**) n=5 dishes totaling 35 neurons (control group), 8 dishes totaling 42 neurons (OVA group), and 10 dishes totaling 76 neurons (OVA + FPM group). p-Values were determined by nested one-way ANOVA with post hoc Bonferroni's. p-Values are shown in the figure.

## Transcriptomic reprogramming of Trpv1[+] neurons

Recent data indicate that the asthma-associated cytokines IL-4 and IL-13 can reprogram lung-innervating nociceptor neurons to adopt a pro-allergic phenotype (*Crosson et al., 2024*). To investigate whether pollution-exacerbated asthma similarly reprograms these neurons, we used *Trpv1*[cre]::*tdTomato*[fl/wt] mice subjected to our standard OVA protocol, with or without FPM. We isolated and dissociated TRPV1[+] JNC neurons, enriched them by excluding satellite glial and immune cells, and then purified them by FACS-sorting followed by RNA-sequencing. Our analysis revealed that

several differentially expressed genes (DEGs) were upregulated in the OVA-FPM group compared to naive mice (*Lifr*, *Oprm1*) (*Figure 2A and B*), in OVA-FPM compared to OVA alone (*Oprm1*, *Nefh*, *P2ry1*, *Prkcb*, *Gabra1*, *Kcnv1*) (*Figure 2C and D*), and in OVA alone compared to naive controls (*Cntn1*, *Piezo1*, *Npy1r*, *Kcna1*) (*Figure 2E and F*). Collectively, these results suggest that pollution-exacerbated asthma both transcriptomically and functionally reprograms lung-innervating nociceptor neurons (*Supplementary file 1*).

## Silencing nociceptor neurons reduces inflammation

To determine whether silencing nociceptor neurons affects pollution-exacerbated asthma, we adapted a previously established neuron-blocking strategy originally developed for pain and itch neurons. This method uses nonselective ion channels (TRPA1 and TRPV1) as a drug entry port for QX-314, a charged form of lidocaine. During inflammation, QX-314 enters sensory fibers, blocking sodium currents and producing a targeted, long-lasting (>9 hr) electrical blockade of nociceptors without impairing immune cell function (*Tochitsky et al., 2021*; *Lee et al., 2019*; *Binshtok et al., 2007*; *Roversi et al., 2022*). Intranasal administration of QX-314 has already been shown to reverse allergic airway inflammation, coughing, mucus metaplasia, and hyperreactivity (*Talbot et al., 2015*; *Lee et al., 2019*; *Aguilar et al., 2024*; *Ito et al., 2024*; *Verzele et al., 2024*).

In our model, an intranasal dose of QX-314 administered at the peak of inflammation (day 16, 5 nmol, 50 µl) significantly reduced neutrophil counts in bronchoalveolar lavage fluid (BALF) of OVA-FPM-exposed mice, restoring them to levels typically seen in asthmatic (OVA-only) mice (*Figure 3A and B*). To validate these findings, we used mice genetically engineered to either retain ($Trpv1^{wt}::Dta^{fl/wt}$; denoted as TRPV1$^{WT}$) or lack ($Trpv1^{cre}::Dta^{fl/wt}$; denoted as TRPV1$^{DTA}$) *Trpv1*-expressing nociceptors (*Balood et al., 2022*; *Wang et al., 2022*; *Crosson and Talbot, 2022*). Compared with controls, TRPV1$^{DTA}$ mice had markedly fewer BALF neutrophils and lung γδ T cells (*Figure 3C–E*), underscoring the critical role nociceptor neurons play in shaping the inflammatory and immune responses in pollution-exacerbated asthma.

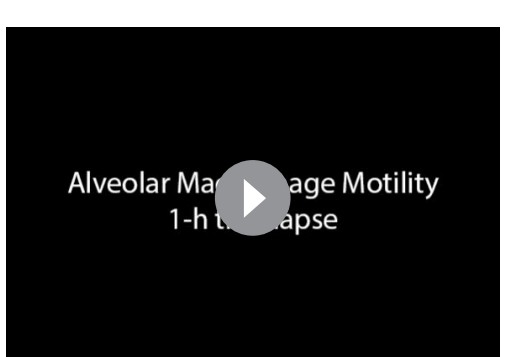

**Video 1.** Intravital recording of alveolar macrophage motility. 6- to 10-week-old male and female littermate control ($Scn10a^{wt}::DTA^{fl/wt}$ denoted as NaV1.8$^{WT}$) and nociceptor-ablated ($Scn10a^{cre}::DTA^{fl/wt}$ denoted as Na$_V$1.8$^{DTA}$) mice were sensitized via intraperitoneal injection of an emulsion containing ovalbumin (OVA; 200 µg/dose) and aluminum hydroxide (1 mg/dose) on days 0 and 7. On day 10, phagocytes were labeled by intranasal injection of PKH26 (25 pmol/dose). Mice were then challenged intranasally with OVA (50 µg/dose) alone or in combination with fine particulate matter (FPM; 20 µg/dose) on days 14–16. Alveolar macrophage intravital imaging was performed on day 17 and is presented as a 1 hr time-lapse video.

https://elifesciences.org/articles/101988/figures#video1

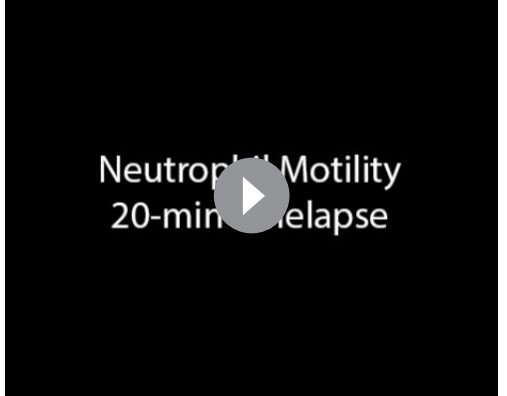

**Video 2.** Intravital recording of neutrophil motility. Male and female littermate control ($Scn10a^{wt}::DTA^{fl/wt}$ denoted as Na$_V$1.8$^{WT}$) and nociceptor-ablated ($Scn10a^{cre}::DTA^{fl/wt}$ denoted as Na$_V$1.8$^{DTA}$) mice (6–10 weeks of age) were sensitized via an intraperitoneal injection of an ovalbumin (OVA; 200 µg/dose) and aluminum hydroxide (1 mg/dose) emulsion on days 0 and 7. On days 14–16, mice were challenged intranasally with OVA (50 µg/dose) alone or in combination with fine particulate matter (FPM; 20 µg/dose). Immediately before intravital imaging on day 17, an intravenous Ly6G antibody was administered to label neutrophils. The resulting recording is presented as a 20 min time-lapse video.

https://elifesciences.org/articles/101988/figures#video2

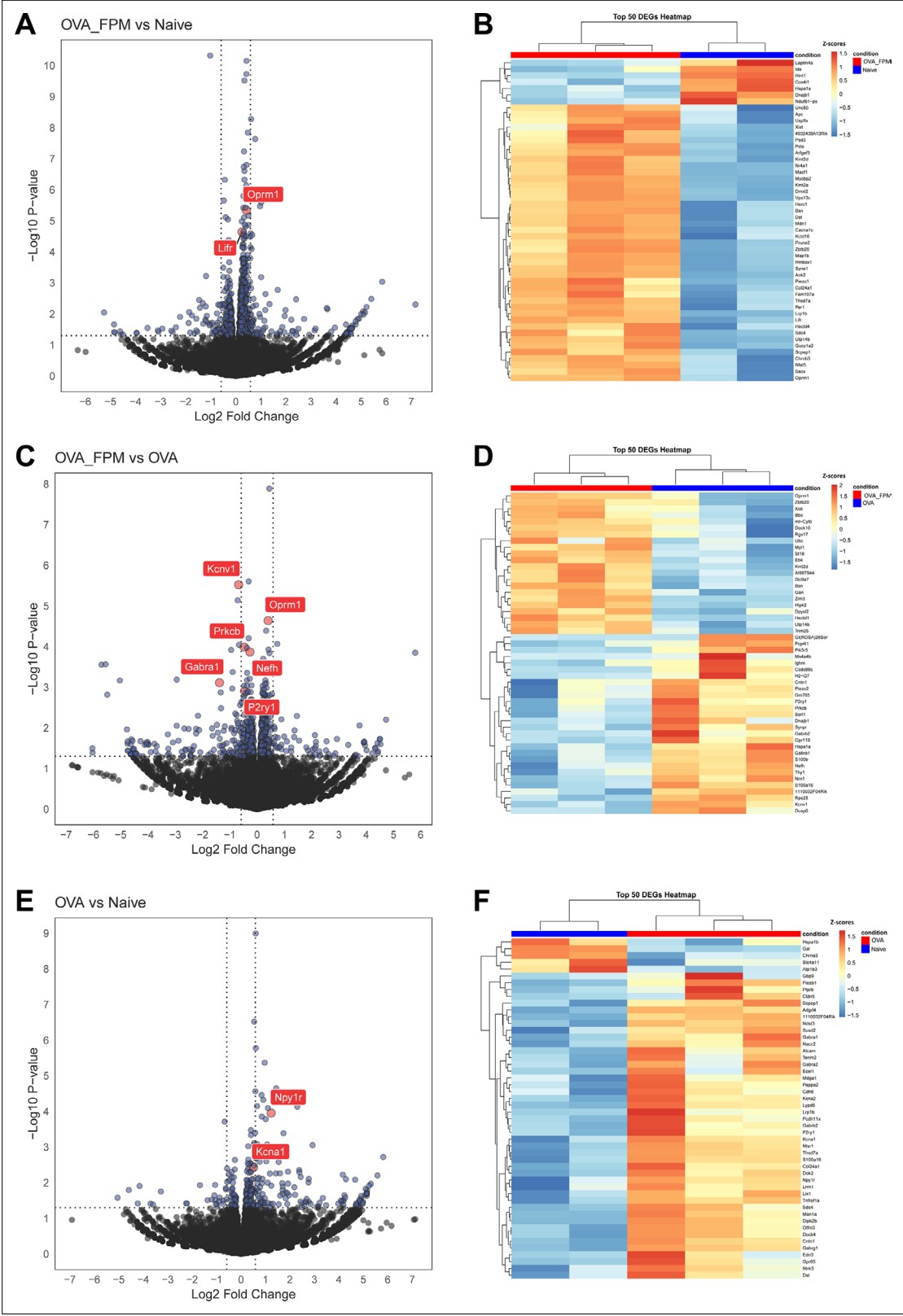

**Figure 2.** Air pollution reprograms the transcriptome of nociceptor neurons. (**A**–**F**) Naïve male and female *Trpv1*^cre^::*tdTomato*^fl/wt^ mice (6–10 weeks of age) underwent either a pollution-exacerbated asthma protocol, the classic OVA protocol, or remained naïve. On day 17 (peak inflammation), jugular-nodose-complex (JNC) neurons were harvested and dissociated, and *Trpv1*^+^ neurons (tdTomato^+^) were sorted via FACS to remove stromal cells and non-peptidergic neurons. RNA was then isolated for sequencing. Volcano plots (**A, C, E**) and heatmaps (**B, D, F**) show differentially expressed genes

*Figure 2 continued on next page*

*Figure 2 continued*

(DEGs) for three comparisons: OVA+FPM vs. naïve (**A–B**), OVA+FPM vs. OVA alone (**C–D**), and OVA alone vs. naïve (**E–F**). Notable genes with increased expression include *Lifr* and *Oprm3* in OVA+FPM vs. naïve, *Oprm1*, *Nefh*, *P2ry1*, *Prkcb*, *Gabra1*, and *Kcnv1* in OVA+FPM vs. OVA, and *Npy1r* and *Kcna1* in OVA alone vs. naïve. Data are presented either as volcano plots (**A, C, E**), showing the $\log_2$ fold change of TPM between groups along with the corresponding $-\log_{10}$ p-values from DESeq2 analysis, or as heatmaps (**B, D, F**), showing the z-scores of rlog-transformed normalized counts. The experimental groups were naïve (n=2; **A–B, E–F**), OVA (n=3; **C–F**), and OVA-FPM (n=3; **A–D**). p-Values were determined by DESeq2 (**A, C, E**) and are indicated in the figure.

## Impact on myeloid cell motility

Building on these observations, we used intravital microscopy (*Kulle et al., 2024*; *Neupane et al., 2020*) to understand how pollution-exacerbated asthma and lung innervation influence the motility of AMs and neutrophils. Although the total number of AMs (labeled by i.n. PKH26) remained unchanged (*Figure 4A and B*), their net displacement was lower in nociceptor-ablated (*Scn10a*^cre^::DTA^fl/wt^; denoted as Na$_V$1.8^DTA^) mice than in littermate controls (*Scn10a*^wt^::DTA^fl/wt^; denoted as Na$_V$1.8^WT^; *Figure 4C–F*, *Video 1*). In contrast, neutrophil accumulation (labeled by i.v. αLy6G) was observed in pollution-exposed, asthmatic littermate controls, but was absent in nociceptor-ablated mice (*Figure 4G and H*). These data are consistent with our BALF flow cytometry data (*Figure 3A and C*). Furthermore, total neutrophil displacement was higher in Na$_V$1.8^DTA^ mice (*Figure 4I–L*, *Video 2*), without any bias toward a specific behavior subtype (e.g. adherent, crawling, patrolling, or tethering; *Figure 4L*). Taken together, these data suggest that nociceptor neurons modulate AM motility and neutrophil accumulation while limiting the motility of recruited neutrophils.

## Elevated cytokines and artemin under pollution-exacerbated asthma

To further explore the role of nociceptor neurons in airway inflammation, we performed an unbiased multiplex cytokine array. This revealed elevated levels of the asthma-promoting cytokine IL-4 and TNFα, with TNFα levels returning to baseline following nociceptor ablation (*Figure 5A*). In addition, targeted enzyme-linked immunosorbent assay (ELISA) showed increased artemin levels, which returned to normal in the absence of nociceptor neurons (*Figure 5B*). Artemin, a member of the glial cell line-derived neurotrophic factor (GDNF) family, is pivotal for the development and function of sympathetic (*Honma et al., 2002*) and sensory (*Ilieva et al., 2019*) neurons through its interaction with the GFRα3-RET receptor complex. Activation of this receptor supports neuronal survival

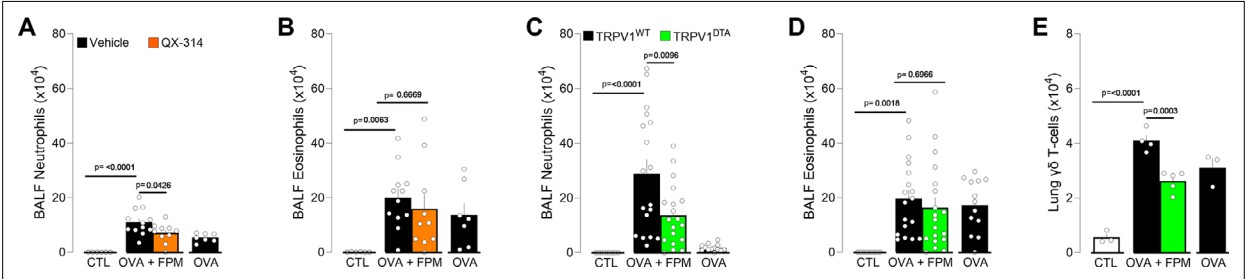

**Figure 3.** Nociceptor neurons control pollution-exacerbated asthma. (**A–B**) Male and female C57BL/6 mice (6–10 weeks of age) were sensitized intraperitoneally with ovalbumin (OVA; 200 µg/dose in 200 µl) and aluminum hydroxide (1 mg/dose in 200 µl) on days 0 and 7. On days 14–16, mice were challenged intranasally with OVA (50 µg/dose in 50 µl) alone or with fine particulate matter (FPM; 20 µg/dose in 50 µl). On day 16, 30 min after the final challenge, mice received intranasal QX-314 (5 nmol/dose in 50 µl). Bronchoalveolar lavage fluid (BALF) was collected on day 17 and analyzed by flow cytometry. Compared with naïve or OVA-exposed mice, those co-challenged with OVA+FPM showed increased BALF neutrophils (**A**). QX-314 treatment normalized these levels, while BALF eosinophil levels remained comparable (**B**). (**C–E**) Male and female littermate control (TRPV1^WT^) and nociceptor-ablated (TRPV1^DTA^) mice (6–10 weeks of age) were sensitized and challenged under the same OVA±FPM protocol (days 0, 7, and 14–16). BALF or lungs were collected on day 17 and assessed by flow cytometry. Compared with naïve or OVA-exposed mice, OVA+FPM co-challenged mice exhibited higher BALF neutrophils (**C**) and lung γδ T cells (**E**). Nociceptor ablation protected against these increases (**C, E**), while BALF eosinophil levels remained comparable (**D**). Data are shown as mean ± SEM (**A–E**). Experiments were replicated twice, and animals pooled (**A–E**). N are as follows: (**A–B**) control (n=6), OVA (n=7), OVA-FPM (n=12), OVA-FPM+QX-314 (n=10), (**C–D**) TRPV1^WT^ + control (n=9), TRPV1^WT^ + OVA (n=13), TRPV1^WT^ + OVA-FPM (n=18), TRPV1^DTA^ + OVA-FPM (n=19), (**E**) TRPV1^WT^ + control (n=3), TRPV1^WT^ + OVA (n=3), TRPV1^WT^ + OVA-FPM (n=4), TRPV1^DTA^ + OVA-FPM (n=5). p-Values were determined by a one-way ANOVA with post hoc Tukey's (**A–E**). p-Values are shown in the figure.

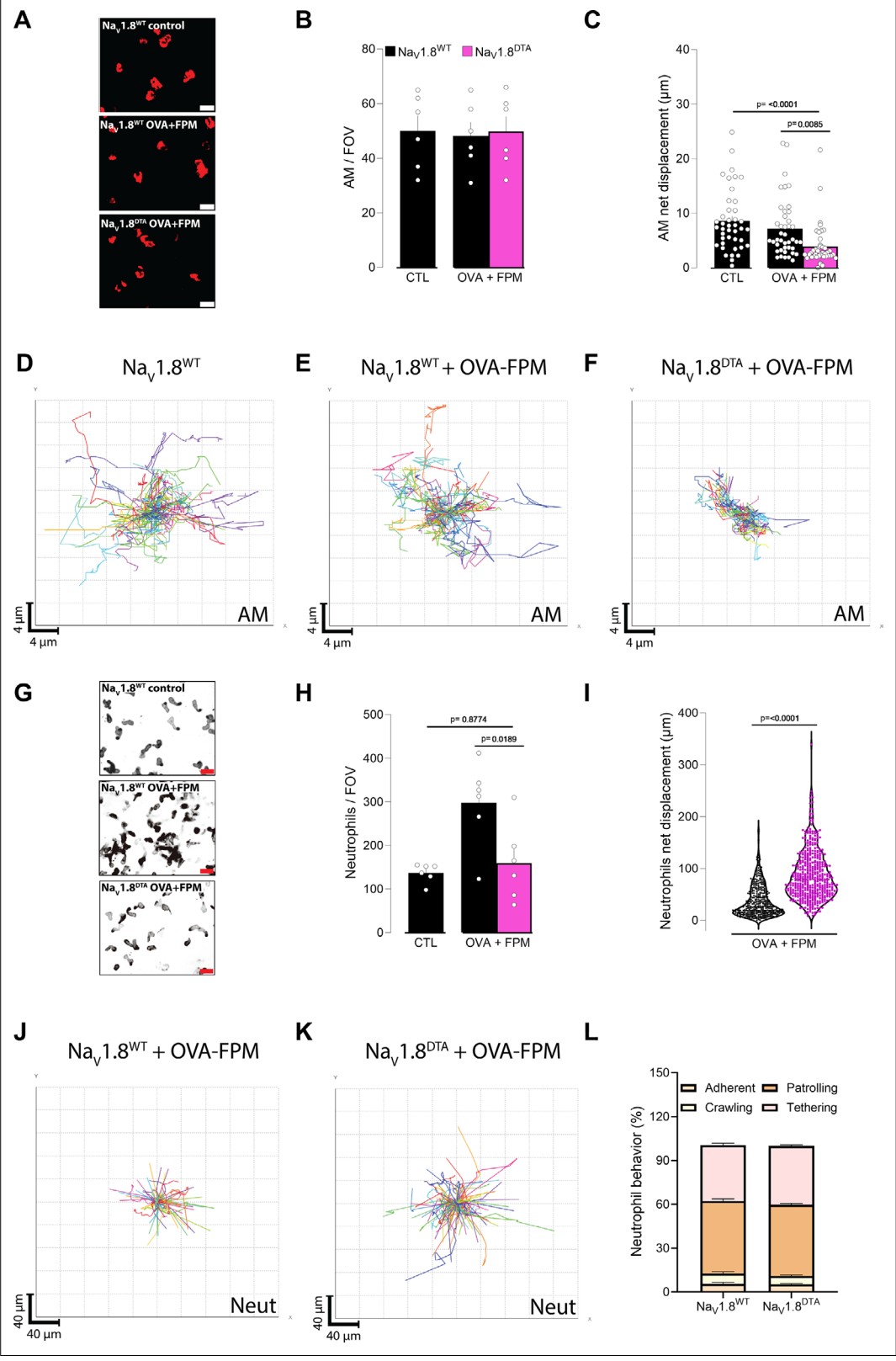

**Figure 4.** Vagal sensory neurons gatekeep alveolar macrophage (AM) motility and neutrophil numbers. (**A–D**) Male and female littermate control (*Scn10a*wt::*Dta*fl/wt denoted as Na$_V$1.8WT) and nociceptor-ablated (*Scn10a*cre::*Dta*fl/wt denoted as Na$_V$1.8DTA) mice (6–10 weeks of age) were sensitized via intraperitoneal injection with an emulsion of ovalbumin (OVA; 200 µg/dose) and aluminum hydroxide (1 mg/dose) on days 0 and 7. Phagocytes were labeled by

*Figure 4 continued on next page*

*Figure 4 continued*

intranasal injection of PKH26 Red Fluorescent Cell Linker Kit (25 pmol/dose) on day 10. Mice were then challenged intranasally with OVA (50 µg/dose) alone or in combination with fine particulate matter (FPM; 20 µg/dose) on days 14–16, and images were acquired on day 17. (**A**) Representative maximum-intensity projection of AMs (red). Scale bar: 20 µm. (**B**) Quantification of AM numbers per field of view (FOV). (**C**) Net displacement of AMs over 1 hr. (**D–F**) Representative tracks of individual AMs (each color represents a single AM) over 1 hr. While the AM numbers (**A–B**) were not impacted, their net displacement (**C–F**) was reduced in OVA-FPM-exposed $Na_V1.8^{DTA}$ mice. (**G–L**) Male and female littermate control ($Scn10a^{wt}::Dta^{fl/wt}$ denoted as $Na_V1.8^{WT}$) and nociceptor-ablated ($Scn10a^{cre}::Dta^{fl/wt}$ denoted as $Na_V1.8^{DTA}$) mice (6–10 weeks of age) were sensitized via intraperitoneal injection with an emulsion of OVA (200 µg/dose) and aluminum hydroxide (1 mg/dose) on days 0 and 7. Mice were then challenged intranasally with OVA (50 µg/dose) alone or in combination with FPM (20 µg/dose) on days 14–16, and images were acquired on day 17. Immediately prior to performing the intravital imaging, we administered an intravenous Ly6G antibody to label neutrophils. (**G**) Representative maximum-intensity projection of neutrophils (black). Scale bar: 20 µm. (**H**) Quantification of neutrophil numbers per FOV. (**I**) The total displacement of neutrophils over 20 min. (**J–K**) Representative tracks of individual neutrophils (each color represents a single neutrophil) over 20 min. (**L**) Frequency of neutrophil behaviors per FOV (adherent, crawling, patrolling, or tethering). Data show that OVA-FPM-exposed littermate control mice present an increase in neutrophil numbers per FOV (**G–H**), an effect absent in $Na_V1.8^{DTA}$. Interestingly, OVA-FPM-exposed $Na_V1.8^{DTA}$ mice show an increase in neutrophil net displacement (**I–K**), an effect irrespective to one of the specific behaviors tested (**L**). Data are presented as representative image (**A, G**; scale bar: 20 µm), mean ± SEM (**B, C, H**), spider plot (**D–F, J–K**), violin plot showing median (**I**), and stacked bar graph showing mean ± SEM (**L**). N are as follows: (**B**) $Na_V1.8^{WT}$ + control (n=6), $Na_V1.8^{WT}$ + OVA-FPM (n=6), $Na_V1.8^{DTA}$ + OVA-FPM (n=6), (**C**) $Na_V1.8^{WT}$ + control (n=42), $Na_V1.8^{WT}$ + OVA-FPM (n=42), $Na_V1.8^{DTA}$ + OVA-FPM (n=42), (**H**) $Na_V1.8^{WT}$ + control (n=6), $Na_V1.8^{WT}$ + OVA-FPM (n=6), $Na_V1.8^{DTA}$ + OVA-FPM (n=6), (**I**) $Na_V1.8^{WT}$ + OVA-FPM (n=385), $Na_V1.8^{DTA}$ + OVA-FPM (n=469), (**J**) $Na_V1.8^{WT}$ + OVA-FPM (n=10), $Na_V1.8^{DTA}$ + OVA-FPM (n=11). p-Values were determined by a one-way ANOVA with post hoc Tukey's (**B, C, H**) or unpaired Student's t-test (**I, L**). p-Values are shown in the figure.

and growth during embryogenesis and drives neuronal hyperactivity under inflammatory conditions (*Lippoldt et al., 2013*; *Elitt et al., 2006*; *Elitt et al., 2008*).

## Mechanistic insights: artemin-nociceptor axis

To determine which vagal neuron subsets express the artemin receptor GFRα3-RET, we performed an in silico analysis of single-cell RNA-sequencing data (*Kupari et al., 2019*; *Zhao et al., 2022*) and found that *Gfra3* is expressed by several nociceptor subtypes (*Figure 5C and D*), including JG3 (*Osmr*-expressing), JG4 (*Trpa1/Sstr2*-expressing), and JG6 (*Trpm8*-expressing). Concurrently, ImmGen data (*Immunological Genome Project, 2020*) indicated robust *Artn* expression in macrophages, with *Ahr* levels slightly lower than those in ILC2s and eosinophils (*Figure 5—figure supplement 1*). We then conducted an in silico analysis of data from *Karlsson et al., 2021*; *Figure 5—figure supplement 2A*, *Uhlén et al., 2005*; *Figure 5—figure supplement 2B*, and *Abdulla et al., 2023* (*Figure 5—figure supplement 2C*), showing that *Ahr* is also expressed in human lung, that AHR protein is detected in human lung macrophages, and that *Artn* and *Ahr* are co-expressed in lung macrophages, respectively. Guided by these findings, we isolated AMs from naive C57BL/6 mice and exposed them to FPM, observing a time-dependent increase in *Artn* transcripts (*Figure 5E–G*). These results highlight key cellular interactions involving FPM, AhR, and artemin in the context of lung inflammation.

A recent study (*Elitt et al., 2006*) linked artemin overexpression to increased mRNA levels of several nociceptor markers (*Gfra3*, *TrkA*, *Trpv1*, *Trpa1*), accompanied by heightened thermal sensitivity. Consistent with this, our experiments in JNC neurons from OVA-FPM co-exposed mice revealed amplified calcium responses to the TRPA1 agonist AITC (*Figure 1B and C*). Follow-up tests in naive C57BL/6 mice showed elevated AITC responsiveness in artemin-exposed JNC neurons compared to vehicle-treated controls (*Figure 5H–J*). Collectively, these data suggest that AMs, through pollutant-driven production of artemin, sensitize nociceptor neurons and thereby potentiate allergic airway inflammation. This pathway underscores a critical link between environmental pollutants and the neurogenic exacerbation of allergic responses (*Figure 6*).

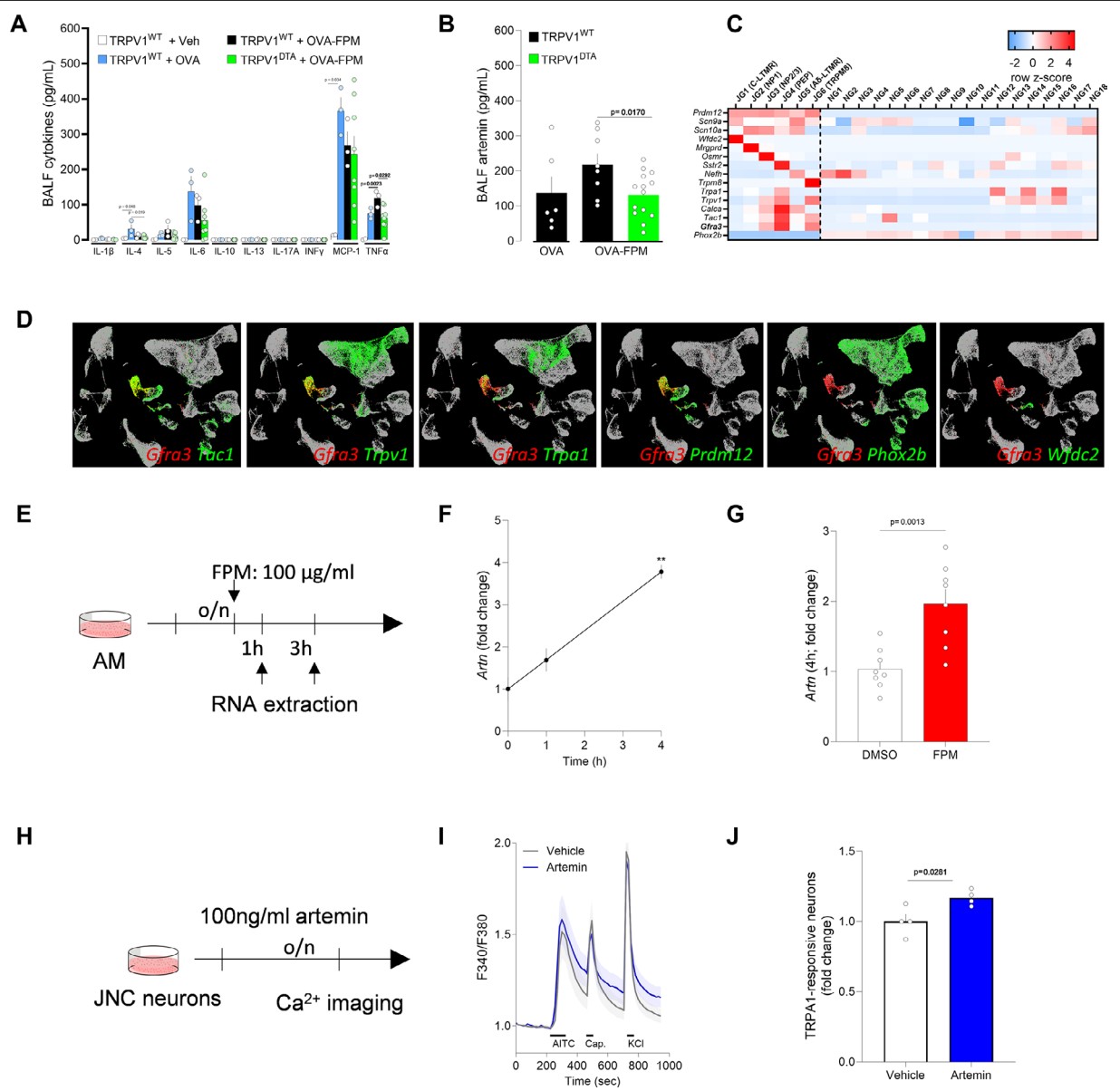

**Figure 5.** Artemin sensitizes TRPA1 activity in vagal sensory neurons. (**A–B**) Male and female littermate control (TRPV1^WT) and nociceptor-ablated (TRPV1^DTA) mice (6–10 weeks of age) were sensitized and challenged under the same ovalbumin (OVA)±fine particulate matter (FPM) protocol (days 0, 7, and 14–16). Bronchoalveolar lavage fluid (BALF) was collected on day 17 and assessed by multiplex array and enzyme-linked immunosorbent assay (ELISA). Compared with naïve or OVA-alone groups, OVA+FPM co-challenged mice exhibited levels of TNFα and artemin. Notably, ablating nociceptors prevented these increases. (**C**) In silico analysis of the GSE124312 dataset (**Kupari et al., 2019**). The heatmap displays transcript expression levels for the pan neural-crest lineage transcription factor (*Prdm12*), voltage-gated sodium channels (*Scn9a, Scn10a*), jugular subset markers (*Wfdc2, Mrgprd, Osmr, Sstr2, Nefh, Trpm8*), peptidergic neuron markers (*Trpa1, Trpv1, Calca, Tac1, Gfra3*), and the pan placodal lineage marker (*Phox2b*). Gfra3 expression is enriched in the peptidergic neuron cluster labeled JG4. Experimental details and cell clustering are described by **Kupari et al., 2019**. (**D**) In silico analysis of GSE192987 (**Zhao et al., 2022**) showing co-expression of *Gfra3* with *Trpa1* and other inflammatory markers. Data are visualized as row z-scores in a heatmap or via UMAPs (TPTT>1). Experimental details and cell clustering are described by **Zhao et al., 2022**. (**E–G**) Alveolar macrophages (3×10^5 cells/well) from naïve male and female C57BL/6 mice were cultured overnight and then stimulated with vehicle (DMSO) or FPM (100 µg/ml). RNA was extracted 1 and 4 hr post-stimulation, and *Artn* expression was assessed using quantitative PCR (qPCR). FPM exposure increased *Artn* transcript levels at both 1 and 4 hr (**F, G**). (**H–J**) Naïve mice jugular-nodose-complex neurons were harvested, pooled, and cultured overnight with either vehicle or artemin (100 ng/ml). Cells were sequentially stimulated with AITC (TRPA1 agonist; 300 µM at 240–270 s), capsaicin (TRPV1 agonist; 300 nM at 320–335 s), and KCl (40 mM at 720–735 s). The percentage of AITC-responsive neurons (among all KCl-responsive cells) was normalized to vehicle-treated controls for each batch of experiments. Artemin-treated neurons showed increased responsiveness to AITC, while responses to capsaicin and KCl were unchanged (**I–J**). Data are presented as means ± SEM (**A–B, F–G, J**), heatmap displaying the z-score of DESeq2 normalized counts (**C**), tSNE plots

*Figure 5 continued on next page*

*Figure 5 continued*

(**D**), schematics (**E, H**), means ± 95% CI of maximum Fura-2AM (F/F$_0$) fluorescence (**I**). N are as follows: (**A**) TRPV1$^{WT}$ + control (n=2), TRPV1$^{WT}$ + OVA (n=3) TRPV1$^{WT}$ + OVA-FPM (n=3), TRPV1$^{DTA}$ + OVA-FPM (n=8), (**B**) TRPV1$^{WT}$ + OVA (n=6) TRPV1$^{WT}$ + OVA-FPM (n=8), TRPV1$^{DTA}$ + OVA-FPM (n=14), (**F**) n=2/time point, (**G**) n=8/group, (**I**) vehicle (n=107 neurons), artemin (n=122 neurons); (**J**) n=4/group. p-Values were determined by a one-way ANOVA with post hoc Tukey's (**A, B**) or unpaired Student's t-test (**G, J**). p-Values are shown in the figure.

The online version of this article includes the following figure supplement(s) for figure 5:

**Figure supplement 1.** In silico analysis of *Artn* expression in mouse immune cells.

**Figure supplement 2.** In silico re-analysis of Ahr and Artn expression in human tissues.

## Discussion

Our research, in line with other studies, indicates that nociceptor neurons promote regulatory immunity in the context of bacterial (*Pinho-Ribeiro et al., 2023*), viral (*Filtjens et al., 2021*), or fungal (*Kashem et al., 2015*) infections, as well as malignancies (*Balood et al., 2022*). To our knowledge, however, the impact of neuro-immunity on neutrophilic asthma is less understood. Using a model of pollution-exacerbated airway inflammation (*Thio et al., 2022*), we found that ablation or silencing of nociceptor neurons prevented the induction of neutrophilic airway inflammation, revealing a potentially novel therapeutic avenue for treating refractory asthma. This result parallels our earlier findings that nociceptor neuron silencing via charged blockers of voltage-gated sodium channels (*Talbot et al., 2015*; *Mathur et al., 2021*; *Tochitsky et al., 2021*) or calcium channels (*Lee et al., 2019*) can halt eosinophilic airway inflammation.

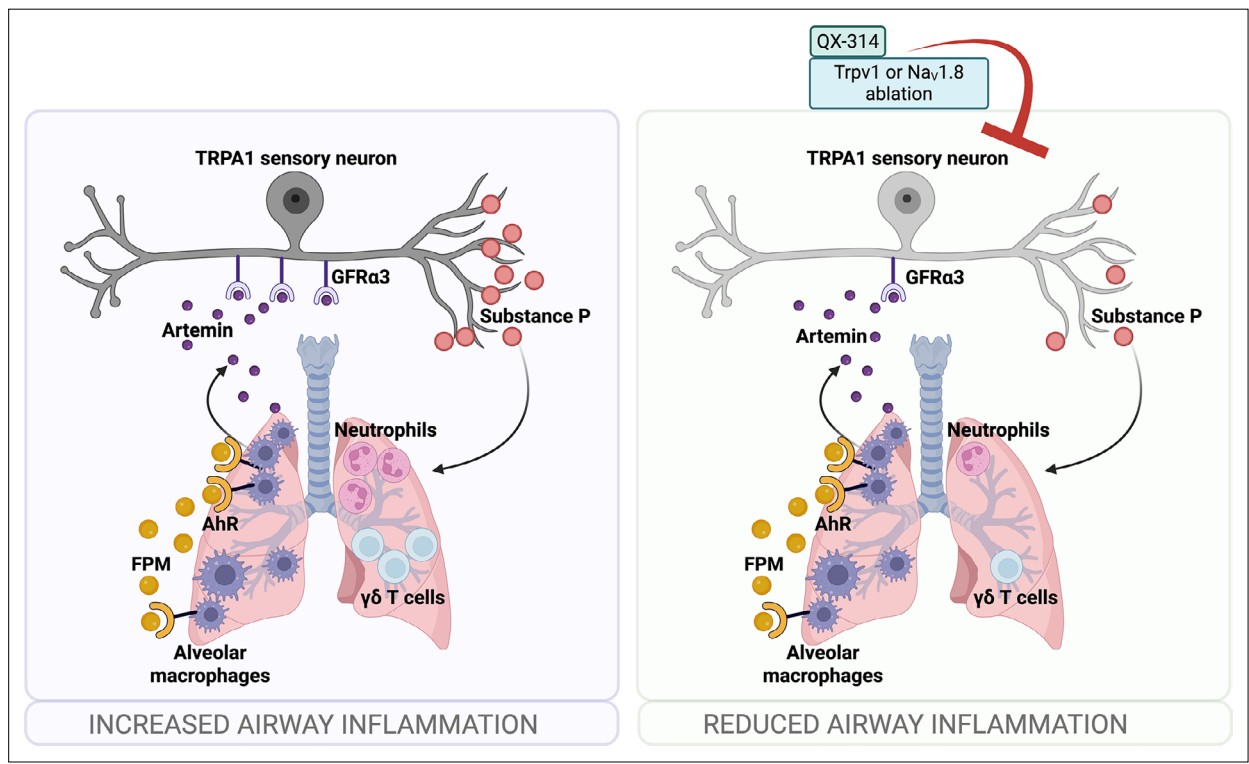

**Figure 6.** Schematic of nociceptor involvement in pollution-exacerbated allergic asthma. In our study, mice were exposed to PM$_{25}$ particles and ovalbumin (OVA) to model pollution-exacerbated asthma. Compared to mice exposed to OVA alone, co-exposure to PM$_{25}$ and OVA significantly increased bronchoalveolar lavage fluid (BALF) neutrophils and lung γδ T cell levels. To counteract this heightened airway inflammation, we administered intranasal QX-314—a charged lidocaine derivative—at the peak of inflammation, effectively normalizing BALF neutrophil levels. Ablation of TRPV1$^+$ nociceptor neurons produced a similar effect. Further analysis with calcium imaging revealed that neurons from the jugular-nodose complex in pollution-exposed asthmatic mice were more sensitive via their TRPA1 channels. Levels of TNFα and the growth factor artemin were also elevated in the BALF of these mice, returning to normal following nociceptor ablation. We identified alveolar macrophages as the source of artemin, which they secrete upon sensing fine particulate matter (FPM) through aryl hydrocarbon receptors. Artemin, in turn, heightened TRPA1 responsiveness to its agonist (mustard oil), thereby exacerbating airway inflammation. Our findings suggest that silencing nociceptor neurons can disrupt this pathway, offering a novel therapeutic approach to mitigate neutrophilic airway inflammation driven by pollution.

During bacterial infections, Pinho-Ribeiro et al. demonstrated that nociceptor neurons inhibit neutrophil influx and antimicrobial activity by releasing the neuropeptide CGRP (*Pinho-Ribeiro et al., 2018*). In addition, Ugolini and colleagues showed that ablating Na$_V$1.8$^+$ nociceptor neurons during HSV-1 infection increases skin neutrophil infiltration, cytokine production, and lesion severity (*Filtjens et al., 2021*). Building on these findings, we used intravital imaging to study neuro-immune interactions in our model. We observed that the total number of AMs remained unchanged in OVA-exposed mice lacking nociceptor neurons, but their displacement was significantly reduced. In contrast, pollution-exposed asthmatic mice lacking nociceptor neurons exhibited a marked decrease in neutrophil recruitment and a corresponding increase in their displacement. These shifts in cellularity occurred regardless of whether nociceptor neurons were genetically ablated (*Trpv1*$^+$ or *Scn10a*$^+$ populations) or pharmacologically silenced with QX-314, as confirmed by flow cytometry. Although individual neutrophil behaviors—such as adherence, crawling, patrolling, and tethering—did not change, neuronal ablation effectively 'removed the brakes' on neutrophil motility. This likely led to reduced neutrophil accumulation in the lung and overall dampening of lung inflammation in pollution-exacerbated asthma. These findings are consistent with the pioneering work by Baral et al., who showed that vagal *Trpv1*$^+$ afferents suppress neutrophil recruitment and modulate lung γδ T cell numbers via CGRP signaling, thus supporting essential antibacterial clearance (*Baral et al., 2018*). Related studies in influenza-infected mice revealed that vagal nociceptor deficiency significantly alters the lung immune landscape—expanding neutrophils and monocyte-derived macrophages (*Yang et al., 2024*)—and transcriptional analyses further indicated disrupted interferon signaling, as well as imbalances among neutrophil subpopulations in nociceptor-ablated mice. Collectively, these results highlight the bidirectional crosstalk between recruited and tissue-resident phagocytes and nociceptors in air-pollution-exacerbated allergic airway inflammation.

The AhR is a crucial modulator of inflammation, as seen in psoriasis models, where its activation reduces inflammation, whereas its absence exacerbates disease (*Bissonnette et al., 2021*; *Di Meglio et al., 2014*). Besides being expressed in AMs, AhR is also present in ILC2s, eosinophils, and neurons. A recent preprint suggests AhR has a dual role in neural protection and axon regeneration: AhR activation in dorsal root ganglion neurons inhibits axon growth (*Halawani, 2024*). At the same time, its deletion lessens inflammation and stress signaling, driving pro-regenerative pathways (*Halawani, 2024*). Additional work indicates that AhR regulates the gut-brain axis (*Barroso et al., 2021*). Although we have not directly examined AhR in nociceptor neurons, our findings suggest that neurons, much like AMs, may sense FPM via AhR. This possibility could explain the direct neuronal transcriptomic reprogramming we observed after pollutant exposure. Ongoing studies aim to probe this hypothesis using nociceptor-specific AhR knockout models.

A growing body of evidence indicates that airway-innervating sensory neurons undergo significant transcriptomic alterations in neutrophilic asthma. Although some lung innervation arises from the dorsal root ganglia—especially for nociception in the upper thoracic segments (T1–T5)—the majority (70–80%, sometimes up to 90%) of afferent fibers originate from the JNC (*Kim, 2022*; *McGovern et al., 2015*). Because the JNC mediates critical mechanosensory and chemosensory functions, our analysis focuses on this primary vagal pathway. While single-cell RNA-sequencing has revealed that JNC neurons are highly heterogeneous and control breathing and tracheal reflexes (*Prescott et al., 2020*; *Nonomura et al., 2017*; *Chang et al., 2015*; *Usoskin et al., 2015*; *Schappe et al., 2024*; *Bin et al., 2023*), the molecular changes that occur during chronic inflammation remain poorly characterized.

Using Na$_V$1.8 reporter mice and intranasal retrograde tracing, we identified a subset of nociceptors innervating the lungs that are reprogrammed in allergic airway inflammation and by the asthma-driving cytokine IL-13 (*Crosson et al., 2024*). Our transcriptomic data using sorted *Trpv1*$^+$ neurons confirm the upregulation of *Npy1r* during allergic airway inflammation (*Crosson et al., 2024*) and highlight the upregulation of *Cacna1c* (*Isensee et al., 2021*) and *Dmxl2* (*Costain et al., 2019*) during pollution-exacerbated asthma, suggesting heightened vesicle trafficking and changes in synaptic release dynamics that may influence neuronal sensitivity and plasticity in asthmatic conditions. Other genes such as *Oprm1* (*Fillingim et al., 2005*) (mu-opioid receptor) and *Hint1* (*Garzón et al., 2015Garzón et al., 2015*) (an opioid signaling modulator) suggest altered pain or irritant sensation in the lung, potentially shaping cough reflexes or bronchospasm. Changes in cytoskeletal regulators, including *Map1b* (*Guo et al., 2023*) and *Apc* (*Duraikannu et al., 2018*), indicate that JNC axons may remodel

their structure and synaptic architecture, potentially altering how airway signals are relayed to central circuits. *Lifr* (*Davis and Pennypacker, 2018*; *Jain et al., 2024*; *Thakur et al., 2014*; *Wang et al., 2024*) (leukemia inhibitory factor receptor alpha), which supports neuron survival and differentiation, may also be important in inflammatory or injury contexts in the lung.

Pathway analysis reveals gene clusters involved in synaptic structure, cytoskeletal organization, and epigenetic regulation. Enrichment in terms such as 'synapse', 'presynaptic active zone', and 'post-synaptic density' indicates that both neurotransmitter release and postsynaptic excitability are being recalibrated, potentially amplifying neuro-immune feedback loops and leading to maladaptive reflexes like excessive bronchoconstriction or persistent cough in neutrophilic asthma. In parallel, pathways related to 'cellular component organization', 'negative regulation of inclusion body assembly', and 'protein quality control' show that neurons may upregulate protective or adaptive mechanisms in response to chronic airway inflammation. These transcriptomic alterations point to a dynamic neuro-immune crosstalk in the inflamed lung, with neurons adapting their excitability and structural integrity in response to cytokines, environmental pollutants, and inflammatory cells, potentially exacerbating or perpetuating the pathophysiology of neutrophilic asthma.

AMs act as a critical early warning system in the lung (*Neupane et al., 2020*), detecting pollutants such as FPM and initiating protective reflexes by secreting artemin. This cytokine activates and sensitizes nociceptor neurons—particularly Trpa1-expressing fibers—to noxious stimuli. Single-cell RNA-sequencing identified a subset of *Trpa1*[+], *Sstr2*[+] neurons that express the GDNF receptor *Gfrα3* and are thereby sensitized by artemin. This finding aligns with earlier data showing that keratinocyte-derived TSLP can sensitize skin-innervating nociceptor neurons, promoting itch and atopic dermatitis (*Wilson et al., 2013*). In the lungs, germline TRPA1 knockout reduces allergic airway inflammation (*Caceres et al., 2009*; *Bessac et al., 2009*), and clinical data from Genentech suggest that TRPA1 agonists are elevated in asthmatic human airways, contributing to both inflammation and hyperreactivity. GDC-0334, a selective TRPA1 antagonist they develop, mitigates airway inflammation, cough, and allergic responses in preclinical models and reduces pain and itch in humans (*Balestrini et al., 2021*). These findings strongly support the central role of TRPA1 sensitization and nociceptor activation in driving asthma pathology.

Our prior work showed that nodose ganglion neurons, which are largely *Trpa1*[+], exhibit heightened thermal sensitivity and outgrowth responses to brain-derived neurotrophic factor but do not respond to nerve growth factor (NGF) (*Crosson and Talbot, 2022*). Here, we demonstrate that vagal nociceptors express GFRα3 and respond to artemin, which sensitizes TRPA1 channels and promotes subsequent airway inflammation. Indeed, JNC neurons undergo substantial transcriptomic reprogramming when exposed to OVA plus FPM, mirroring the effects of pro-asthmatic cytokines. Parallel data from other disease models support this link between inflammation-induced sensitization and elevated artemin. For example, increased *Artn* and *Egr1* levels in atopic dermatitis correlate with enhanced nerve density and scratching, both of which are attenuated in EGR1-deficient mice (*Yeo et al., 2024*). Similarly, artemin overexpression boosts sensory neuron thermal responses, underscoring its importance in driving neuronal outgrowth and sensitivity in atopic conditions (*Elitt et al., 2006*). These observations raise the possibility that blocking GFRα3 may offer a novel strategy to prevent maladaptive nociceptor involvement in pollution-exacerbated asthma.

Our previous work showed that vagal nociceptor neurons can detect allergen-IgE immune complexes and respond by releasing substance P (SP) and VIP, but not CGRP (*Crosson et al., 2021*). These findings were corroborated by analyzing BALF from asthmatic mice (*Crosson et al., 2024*). Moreover, we demonstrated that SP drives mucus metaplasia in asthmatic lungs (*Talbot et al., 2020*) and influences antibody class switching in B cells (*Mathur et al., 2021*), a result that other groups have replicated (*Tynan et al., 2024*; *Aguilar et al., 2024*; *Helme et al., 1987*; *Sponchiado et al., 2021*). We also observed that *Trpv1*[+] nociceptor ablation impairs antigen trafficking to lymph nodes and reduces IgG production (*Tynan et al., 2024*, *Hanes et al., 2016*, *Huang et al., 2021*), possibly decreasing the severity or onset of allergic responses. Emerging work highlights the varied immuno-modulatory roles of different neuropeptides. For instance, CGRP impairs dendritic cell migration in psoriasis (*Hanč et al., 2023*; *Riol-Blanco et al., 2014*), whereas SP enhances dendritic cell migration to lymph nodes in atopic dermatitis (*Perner et al., 2020*). VIP and neuromedin U boost pro-asthmatic cytokines from lung ILC2s (*Talbot et al., 2015*; *Klose et al., 2017*; *Cardoso et al., 2017*; *Wallrapp et al., 2017*; *Nussbaum et al., 2013*), while CGRP can have similar (*Sui et al., 2018*; *Ike et al., 2023*)

or opposing effects (*Wallrapp et al., 2019*). In lung infections, CGRP reduces neutrophil and γδ T cell infiltration and protects against *Staphylococcus aureus* pneumonia (*Baral et al., 2018*; *Huang et al., 2023*). Conversely, CGRP exacerbates psoriasis by inducing dendritic cells to release IL-23, which activates IL-17-producing γδ T cells, intensifying inflammation (*Kashem et al., 2015*). Likewise, during *Candida albicans* infection, sensory neurons release CGRP that stimulates IL-23 production in CD301b⁺ dendritic cells, triggering $T_h17$ and γδ T cell responses (marked by IL-17A and IL-22), enhancing host defense (*Cohen et al., 2019*). In our model, nociceptor neuron ablation diminishes γδ T cell activation, which we posit is chiefly driven by SP/VIP, given the absence of heightened CGRP release in our asthma models (*Talbot et al., 2015*; *Mathur et al., 2021*). Future work will investigate whether neurons directly regulate γδ T cells and neutrophils through these or other neuropeptides.

Previous reports show that *Trpv1*⁺ neuron activation elicits a local type 17 immune response that enhances host defense against *C. albicans* (*Kashem et al., 2015*; *Cohen et al., 2019*) and *S. aureus* (*Cohen et al., 2019*). While our mixed OVA-FPM was expected to induce a strong $T_h17/T_h2$ component, we did not observe an increase in IL-17A in our cytokine bead array. Similarly, increases in classic asthma-driving cytokines were modest (IL-4, IL-5) or absent (IL-13). One possible explanation is that these experiments were conducted in C57BL/6 mice, rather than BALB/c mice (*Thio et al., 2022*). We also noted an increase in TNFα and artemin, both of which were reduced by nociceptor neuron silencing. While we found that TNFα can reprogram nociceptor neurons (*Crosson et al., 2024*), we propose that this TNFα may originate from activated neutrophils (*Giambelluca et al., 2014*; *Feiken et al., 1995*), consistent with the reduced TNFα and neutrophil activation seen in neuron-ablated mice. It is worth noting that the $PM_{25}$ used in this study—Standard Reference Material (SRM) 2786—was obtained from an air intake system in the Czech Republic and that we cannot rule out that it may contain trace levels of LPS that could contribute to this neutrophil-TNFα hypothesis. Our rationale for choosing SRM2786 was that it is commercially available and represents a broad spectrum of ambient air pollutants, in contrast to more specialized sources such as diesel exhaust particles. Future studies will need to validate these findings using wildfire-collected particles (*Kobziar et al., 2022*), LPS-free air intake particles, and replication in BALB/c mice.

In summary, our study highlights how AMs, via AhR-dependent pollutant detection and artemin secretion, sensitize TRPA1⁺ nociceptor neurons, thereby amplifying neutrophilic asthma. We propose several potential therapeutic strategies to curb neutrophilic airway inflammation: (i) targeting AhR signaling in AMs, (ii) blocking artemin's interaction with GFRα3, (iii) using TRPA1 antagonists such as GDC-0334, and (iv) silencing nociceptor neurons with charged lidocaine derivatives. Collectively, these approaches offer new directions to manage the increasingly prevalent yet treatment-resistant, neutrophilic variant of asthma.

# Materials and methods

### Key resources table

| Reagent type (species) or resource | Designation | Source or reference | Identifiers | Additional information |
|---|---|---|---|---|
| Strain, strain background (*Mus musculus*) | C57BL/6J | The Jackson lab | Stock No: 000664; RRID:IMSR_JAX:000664 | Wild-type background strain used for experiments and/or breeding. |
| Strain, strain background (*Mus musculus*) | DTAfl/fl (floxed diphtheria toxin A line; 'DTAfl') | The Jackson lab | Stock No: 010527; RRID:IMSR_JAX:010527 | Cre-dependent DTA expression for genetic ablation when crossed to Cre driver lines. |
| Strain, strain background (*Mus musculus*) | DTAfl/fl (floxed diphtheria toxin A line; 'DTAfl') | The Jackson lab | Stock No: 009669; RRID:IMSR_JAX:009669 | Cre-dependent DTA expression for genetic ablation when crossed to Cre driver lines. |
| Strain, strain background (*Mus musculus*) | tdTomatofl/fl (Ai14; 'tdTomatofl') | The Jackson lab | Stock No: 007914; RRID:IMSR_JAX:007914 | Cre-dependent tdTomato reporter used to label/sort Trpv1⁺ neurons (Trpv1cre/wt::tdTomatofl/wt). |
| Strain, strain background (*Mus musculus*) | TRPV1cre/cre | The Jackson lab | Stock No: 017769; RRID:IMSR_JAX:017769 | Trpv1 promoter-driven Cre; used to target Trpv1⁺ nociceptors for reporter labeling and/or DTA-mediated ablation. |

*Continued on next page*

*Continued*

| Reagent type (species) or resource | Designation | Source or reference | Identifiers | Additional information |
|---|---|---|---|---|
| Strain, strain background (*Mus musculus*) | NaV1.8cre/cre | The Jackson lab | Stock No: 036564; RRID:IMSR_JAX:036564 | NaV1.8 (Scn10a) promoter-driven Cre; used to target NaV1.8+ nociceptors for DTA-mediated ablation. |
| Antibody | Anti-mouse CD45 [clone 30-F11] (rat monoclonal) | BioLegend | Cat# 103128; RRID:AB_493715 | 1:200 to 1:400 |
| Antibody | Anti-mouse CD90.2 (Thy1.2) [clone 53–2.1] (rat monoclonal) | BioLegend | Cat# 140307; RRID:AB_10643585 | 1:200 to 1:400 |
| Antibody | Anti-mouse CD11b [clone M1/70] (rat monoclonal) | BioLegend | Cat# 101243; RRID:AB_2561373 | 1:200 to 1:400 |
| Antibody | Anti-mouse CD11c [clone N418] (Armenian hamster mAb) | BioLegend | Cat 117303; RRID:AB_313772 | 1:200 to 1:400 |
| Antibody | Anti-mouse Ly6C [clone HK1.4] (rat monoclonal) | BioLegend | Cat# 128004; RRID:AB_1236553 | 1:200 to 1:400 |
| Antibody | Anti-mouse Ly6G [clone 1A8] (rat monoclonal) | BioLegend | Cat# 127610; RRID:AB_1134159 | 1:200 to 1:400 |
| Antibody | Anti-mouse Siglec-F [clone 1RNM44N] (rat monoclonal) | Invitrogen | Cat# 12-1702-82; RRID:AB_2637129 | 1:200 to 1:400 |
| Antibody | Anti-mouse TCR γ/δ [clone GL3] (Armenian hamster mAb) | BioLegend | Cat# 118128; RRID:AB_2562771 | 1:200 to 1:400 |
| Antibody | Anti-mouse TCR β [clone H57-597] (Armenian hamster mAb) | BioLegend | Cat# 109201; RRID:AB_313424 | 1:200 to 1:400 |
| Antibody | Anti-mouse CD19 [clone 1D3] (rat monoclonal) | BD Biosciences | Cat# 553783; RRID:AB_395047 | 1:200 to 1:400 |
| Antibody | Anti-mouse NK1.1 [clone PK136] (mouse monoclonal) | Invitrogen | Cat# 25-5941-82; RRID:AB_469665 | 1:200 to 1:400 |
| Antibody | Anti-mouse F4/80 [clone BM8] (rat monoclonal) | Invitrogen | Cat# 14-4801-82; RRID:AB_467558 | 1:200 to 1:400 |
| Antibody | Anti-mouse FcεRIα [clone MAR-1] (rat monoclonal) | BioLegend | Cat# 134318; RRID:AB_10640122 | 1:200 to 1:400 |

## Animals

All procedures involving animals adhered to the guidelines of the Canadian Council on Animal Care (CCAC), McGill University (Mgcl-8184), and the Queen's University Animal Care Committee (UACC, protocol 2384). Mice were housed in individually ventilated cages with free access to water and food under 12 hr light cycles.

Parental strains C57BL/6 (# 000664), *Dta*fl/fl (# 010527, # 009669), *tdTomato*fl/fl (# 007914), *Trpv1*cre/cre (# 017769), and *Scn10a*cre/cre (# 036564) were purchased from The Jackson Laboratory. Male and female mice were bred in-house and used at 6–12 weeks of age. Crosses were performed to generate the following genotypes: *Scn10a*cre/wt::*Dta*fl/wt, *Trpv1*cre/wt::*Dta*fl/wt, *Trpv1*cre/wt::*tdTomato*fl/wt, and litter-mate control, namely *Scn10a*wt/wt::*Dta*fl/wt, *TRPV1*wt/wt::*Dta*fl/wt. Male and female progeny mice were used between 8 and 16 weeks of age.

## OVA model of allergic airway inflammation

On days 0 and 7, mice were sensitized by i.p. injections of 200 µl of a solution containing 1 mg/ml grade V OVA (Sigma, #A5503) and 5 mg/ml aluminum hydroxide (Sigma, #239186) in PBS (Thermo Fisher, #10010023). On days 14–16, mice were anesthetized with isoflurane (2.5%; CDMV, #108737) and intranasally instilled daily with 50 µg OVA in 50 µl PBS with or without FPM (20 µg/dose; NIST

2786). Control mice were sensitized but did not undergo challenges. Unless otherwise indicated, mice were sacrificed on day 17.

## Neuron silencing

In some experiments, QX-314 (Tocris, #1014; 5 nmol in 50 µl) or PBS was administered intranasally on day 16 to control asthmatic mice (as detailed in the asthma protocol). The mice were euthanized on day 17, and BALF, lung tissues, and the jugular-nodose complex were collected.

## Bronchoalveolar lavage

BALF was harvested in anesthetized mice, following a tracheal incision, by lavaging twice with 1 ml of either PBS or FACS buffer (2% FBS and 1 mM EDTA in PBS) through a Surflo ETFE IV Catheter 20G×1″ (Terumo Medical Products, # SR-OX2025CA). The lavage fluid was centrifuged at 350×$g$ for 6.5 min, and the supernatant was collected for ELISA analysis. The cell pellet was resuspended, subjected to RBC lysis (Cytek, # TNB-4300-L100 or Gibco, # A1049201), and stained for surface markers before flow cytometry.

## Flow cytometry

Single-cell suspensions derived from BALF or lung samples were stained with Ghost Dye Violet 510 (Cytek, # 13-0870-T100) and appropriate antibody cocktails in PBS. Cells were incubated at 4°C for 30 min, then fixed with 10% neutral buffered formalin (Sigma-Aldrich, # HT501128) at room temperature for 15 min before data acquisition. To assess eosinophil and neutrophil infiltration in BALF, cells were stained with fluorochrome-conjugated antibodies against CD45 (clone: 30-F11), CD90.2 (clone: 53-2.1), CD11b (clone: M1/70), CD11c (clone: N418), Ly6C (clone: HK1.4), Ly6G (clone: 1A8), and Siglec-F (clone: 1RNM44N). For γδ T cell analysis in lung tissue, staining included CD45 (clone: 30-F11), TCRγδ (clone: GL3), CD90.2 (clone: 53-2.1), and lineage markers TCRβ (clone: H57-597), CD19 (clone: 1D3/CD19), NK1.1 (clone: PK136), CD11b (clone: M1/70), CD11c (clone: N418), F4/80 (clone: BM8), and FcεRIα (clone: MAR-1), obtained from BioLegend or Thermo Fisher Scientific. Data were acquired using a BD FACS Canto II system.

## Lung tissue harvesting

After diaphragm incision and transcardial perfusion with 10 ml of PBS, lung tissues were dissected, minced with razor blades, and either placed in TRIzol Reagent (Invitrogen, # 15596026) for RNA extraction or transferred into a digestion buffer consisting of 1.6 mg/ml collagenase type 4 (Worthington LS004189) and 100 µg/ml DNase I (Roche, # 11284932001) in supplemented DMEM (*Talbot et al., 2011*). The tissues were digested for 45 min at 37°C with mechanical dissociation through 18-gauge needles after 30 min, followed by filtration through a 70 µm nylon mesh and RBC lysis. Cells were resuspended in FACS buffer for flow cytometry or fluorescence-activated cell sorting or in FBS-supplemented DMEM for in vitro stimulation in 96-well plates at 37°C with 5% $CO_2$, after which supernatants were collected.

## AM culture

AMs were obtained from the BALF of naïve mice, where they represented approximately 95% of the recovered cells. After centrifugation and RBC lysis, these cells were seeded at $3×10^5$ per well in 96-well plates containing DMEM (Gibco 11965092) supplemented with 1 mM sodium pyruvate (Gibco, # 11360070), 2 mM GlutaMAX (Gibco, # 35050061), 100 U/ml penicillin, and 100 µg/ml streptomycin (Corning, # 30-002 CI), 10 mM HEPES (Gibco, # 15630080), and 10% FB Essence (VWR 10805-184), and cultured overnight. They were then stimulated with 100 µg/ml FPM (NIST, # 2786) for 1–4 hr, followed by RNA extraction for quantitative PCR (qPCR) analysis.

## Neuron culture

The jugular-nodose complex (JNC) was collected from anesthetized mice following exsanguination and placed in a digestion buffer containing 1 mg/ml (325 U/ml) collagenase type 4 (Worthington, # LS004189), 2 mg/ml (1.8 U/ml) Dispase II (Sigma, # 04942078001), and 250 µg/ml (735.25 U/ml) DNase I (Roche, # 11284932001) prepared in supplemented DMEM without FB Essence. This mixture was incubated at 37°C for 60 min to ensure enzymatic digestion, followed by mechanical dissociation

through progressive pipetting with tips of decreasing diameter and final passage through a 25-gauge needle. The cell suspension underwent density gradient centrifugation at 200×$g$ for 20 min at low acceleration and deceleration, layering 150 mg/ml bovine serum albumin (BSA; Hyclone, # SH30574.02) in PBS to separate the cells. The bottom fraction was collected, RBC lysed, and seeded onto glass-bottom dishes (Abidi, # 81218) coated with 50 µg/ml laminin (Sigma, # L2020) and 100 µg/ml poly-D-lysine (Sigma, # P6407). Cells were cultured overnight in Neurobasal-A medium (Gibco, # 10888022) supplemented with 1 mM sodium pyruvate (Gibco, # 11360070), 2 mM GlutaMAX (Gibco, # 35050061), 100 U/ml penicillin, 100 µg/ml streptomycin (Corning, # 30-002 CI), 10 mM HEPES (Gibco, # 15630080), B-27 supplement (Gibco 17504-044), 50 ng/ml mouse NGF (Gibco, # 13257-019), 2 ng/ml mouse glial-derived neurotrophic factor (GDNF; Novus, # NBP2-61336), and cytosine-β-D-arabinofuranose (Thermo Scientific, # J6567106). In some experiments, 100 ng/ml artemin or 50 µM HCl (as vehicle control) were added in vitro in place of NGF or GDNF. This culture system was subsequently used for calcium imaging.

## Real-time qPCR

qPCR was performed on stimulated AMs that were lysed using TRIzol Reagent and stored at –80°C until RNA extraction. RNA from sorted cells was extracted using the PureLink RNA Micro Scale Kit (Thermo Fisher, # 12183016). In contrast, RNA from lung tissues or lung cell suspensions was extracted with the E.Z.N.A. Total RNA Kit I (Omega Bio-tek, # R6834). Extraction procedures followed the manufacturer's instructions, including phenol-chloroform purification and mixing with an equal volume of isopropanol. Complementary DNA (cDNA) was synthesized using the SuperScript VILO Master Mix (Invitrogen, # 11755050), with 1–2 µg of RNA as the template in each reaction. qPCR was carried out with PowerUp SYBR Green Master Mix (Applied Biosystems, # A25742), using 50–100 ng of cDNA and 200 nM of each primer. The reactions were run on either a Mic qPCR Cycler (Bio Molecular Systems) or a CFX Opus Real-Time PCR System (Bio-Rad Laboratories). The primer pair for *Artn* was Forward: 5′-TGATCCACTTGAGCTTCGGG-3′ and Reverse: 5′-CTCCATACCAAAGGGGTCCTG-3′.

## Calcium imaging recording

Cultured neurons were loaded with 5 µM Fura-2 AM (Cayman Chemical Company, # 34993) and incubated at 37°C for 40 min. After incubation, the cells were washed four times with standard external solution (SES; Boston BioProducts, # C-3030F) and maintained in this solution during imaging. The Fura-2 signals were recorded and used for downstream analyses. Agonists, diluted in SES, were delivered using a ValveLink 8.2 system (Automate Scientific) equipped with 250 µm Perfusion Pencil tips (Automate Scientific) and controlled by Macro Recorder (Barbells Media, Germany). Between drug injections, SES flow was maintained to ensure a complete washout of each agonist. Imaging for Fura-2 experiments was conducted with a NIKON ECLIPSE Ti2 Inverted Microscope using an S Plan Fluor ELWD 20× objective lens to optimize UV light transmission. Images were captured every 3 or 4 s using sCMOS cameras such as PCO.Edge 4.2 LT (Excelitas Technologies), Prime 95B (Teledyne Photometrics), or Orca Flash 4.0 v2 (Hamamatsu Photonics). Regions of interest were manually delineated in NIS-Elements (Nikon), and the F340/F380 ratios were exported to Excel (Microsoft 365) for further analysis (*Pereira et al., 2015*). The data were compressed by calculating a maximum value every 15 s for all subsequent evaluations.

## Intravital microscopy

6- to 10-week-old male and female littermate control (*Scn10a*[wt]::*Dta*[fl/wt] denoted as Na$_V$1.8[WT]) and nociceptor-ablated (*Scn10a*[cre]::*Dta*[fl/wt] denoted as Na$_V$1.8[DTA]) mice were sensitized via i.p. injection of an emulsion containing OVA (200 µg/dose) and aluminum hydroxide (1 mg/dose) on days 0 and 7. On day 10, phagocytes were labeled by intranasal injection of PKH26 Red Fluorescent Cell Linker Kit (Sigma, # PKH26PCL) at 25 pmol/dose in Diluent B. Mice were then challenged intranasally with OVA (50 µg/dose) alone or in combination with FPM (20 µg/dose) on days 14–16. AM intravital imaging was performed on day 17 and is presented as a 1 hr time-lapse video.

Lung images were acquired using a Nikon CSU-X1 multichannel spinning-disk confocal upright microscope with a protocol adapted from previously published methods (*Kulle et al., 2024*; *Neupane et al., 2020*). Mice were anesthetized with 10 mg/kg xylazine hydrochloride and 200 mg/kg ketamine hydrochloride, and body temperature was maintained at 37°C with a heating pad (World Precision

Instruments). The right jugular vein was cannulated for additional anesthetic as needed and for injecting anti-mouse Ly6G Alexa Fluor 647 (BioLegend, clone: 1A8, # 127610) to label neutrophils. After exposing the trachea and inserting a catheter connected to a small rodent ventilator (Harvard Apparatus), the mouse was placed in a right lateral decubitus position. A small incision was made between ribs 4 and 5 to create an opening of about 1.5 cm, and an intercostal lung window was carefully fitted and stabilized by a vacuum of roughly 20 mmHg. Time-lapse images were acquired without delay using a 20× water-immersion objective (numerical aperture 1).

Images are presented as maximum-intensity projections of z-stacks. AM movement was tracked for over 1 hr, and neutrophil behavior was recorded for 20 min. Cell displacement was quantified using the ICY software's manual tracking plugin. Neutrophil behavior was classified into adherent, tethering, crawling, or patrolling using Imaris (Oxford Instruments) spot tracking. Track Duration and Track Speed Mean were used to define each behavior per field of view (FOV). Tethering was determined by a Track Duration under 150 s for cells that rapidly entered the FOV before arresting and exiting again. Adherent cells remained immobile for more than 150 s, with a Track Speed Mean of ≤0.03 μm/s. Crawling cells were motile with steady movement that persisted for at least half of the video's duration; these had a Track Speed Mean >0.03 μm/s and a Track Duration >600 s. Patrolling cells shared the rapid entry observed with tethering cells but, instead of briefly arresting, continued crawling and exited the FOV, showing a Track Speed Mean >0.03 μm/s and a Track Duration >150 s but <600 s.

## Bulk RNA-sequencing

*Trpv1*[cre/wt]::*tdTomato*[fl/wt] mice were sensitized via i.p. injection with a mix of grade V OVA (200 μg/dose; Sigma-Aldrich A5503) and Imject Alum (1 mg/dose; Thermo Fisher 77161) on days 0 and 7. Subsequently, they underwent intranasal challenges with OVA (50 μg/dose), with or without FPM (20 μg/dose; NIST 2786), from days 14 to 16. Control mice were sensitized but not challenged. The mice were euthanized on day 17, JNCs were collected, and *tdTomato*[+] cells from naive, OVA-challenged, and OVA-FPM co-exposed mice were sorted by FACS (*Crosson et al., 2024*), and total RNA was extracted following established protocols. Library preparation was carried out at the Institut de Recherche en Cancérologie et en Immunologie (IRIC) of the Université de Montréal. RNA quality was assessed using an Agilent Bioanalyzer, ensuring a minimum RNA Integrity Number (RIN) of 7.5. Libraries were prepared using a poly(A)-enrichment, single-stranded RNA-sequencing strategy (Kapa-Biosystems, KAPA RNA Hyperprep Kit, #KR1352) and sequenced on an Illumina NextSeq500 platform with 75-cycle single-end reads.

Base calling was performed using Illumina RTA 2.4.11, and demultiplexing was conducted with bcl2fastq 2.20, allowing for one mismatch in the index. Trimmomatic was used to remove adapter sequences and low-quality bases from the 3' end of each read. The remaining high-quality reads were aligned to the GRCm38 mouse genome using STAR v2.5.11, which also generated gene-level read counts. Differential expression analysis was performed using DESeq2 on these read counts, normalized by the DESeq2 pipeline. $\log_2$ fold changes and $-\log_{10}$ p-values were calculated from the normalized data, and genes were considered differentially expressed if their adjusted p-value (false discovery rate) was below 0.05. Further data analysis and visualization were conducted in RStudio. Bulk RNA-sequencing raw and processed data have been deposited in the NCBI's gene expression omnibus (GSE298583).

## Enzyme-linked immunosorbent assay

ELISA was used to measure artemin levels in BALF (R&D Systems, # DY1085-05) following the manufacturer's instructions. Inflammatory cytokines in BALF were detected with a Cytometric Bead Array Flex Set from BD Biosciences: Master Buffer Set (# 558266), IL-1β (# 560232), IL-4 (# 558298), IL-5 (# 558302), IL-6 (# 558301), IL-10 (# 558300), IL-13 (# 558349), IL-17A (# 560283), IFNγ (# 558296), MCP-1 (# 558342), and TNF (# 558299), also used according to the manufacturer's guidelines.

## In silico analysis of mouse immune cells expression profile using the Immgen database

Using the publicly available Immgen database (*Immunological Genome Project, 2020*), we proceed to an in silico analysis of RNA-sequencing data (DESeq2 data) of various mouse immune cells. As per Immgen protocol, RNA-sequencing reads were aligned to the mouse genome GENCODE GRCm38/

mm10 primary assembly (GenBank assembly accession GCA_000001635.2) and gene annotations vM16 with STAR 2.5.4a. The ribosomal RNA gene annotations were removed from the general transfer format file. The gene-level quantification was calculated by featureCounts. Raw reads count tables were normalized by median of ratios method with DESeq2 package from Bioconductor and then converted to GCT and CLS format. Samples with less than 1 million uniquely mapped reads were automatically excluded from normalization. Experimental details are defined here.

## In silico analysis of RNA-sequencing data

In silico analysis of RNA-sequencing data involved extracting information from *Kupari et al., 2019*'s supplementary materials (GSE124312) (), with clusters based on that publication's designations. Additional data from *Zhao et al., 2022* (GSE192987) were re-analyzed using R and plotted with UMAP to visualize the co-expression of relevant genes. Bulk JNC sequencing datasets were analyzed with DESeq2.

## In silico analysis of lung cancer patients' tumor expression profile using single-cell RNA-sequencing

We performed an in silico analysis of single-cell RNA-sequencing data from mouse lung tumor-infiltrating CD45[+] cells in non-small cell lung cancer (NSCLC) models (GSE127465) (*Zilionis et al., 2019*), accessed via the publicly available Broad Institute Single-Cell Portal. The expression levels of *Ahr, Artn, Calca, Vip, Tac1,* and *Trpa1* were plotted within CD45[+] myeloid cell populations using RStudio. Additionally, we conducted a Spearman correlation analysis to explore relationships among these genes. Gene expression data are presented as normalized counts per 10,000. We extracted the relevant expression values and used RStudio to generate dot plots for individual cells in each myeloid population, as well as to perform correlation analyses of the selected genes. Experimental and clustering details can be found at *Zilionis et al., 2019*.

## In silico analysis of human lung-tissue gene expression using the HPA Single Cell Type Atlas

We retrieved the lung dataset generated by *Karlsson et al., 2021*, from the Human Protein Atlas (HPA) Single Cell Type Atlas, which pools single-cell RNA-sequencing reads into annotated clusters and reports expression as per-gene z-scores after median-of-ratios normalization. Using R (v4.3.2), we downloaded the cluster-level expression matrix, isolated all clusters, and extracted transcript values for *Gfra3, Ahr, Artn, Calca, Tac1, Vip,* and *Trpa1*. The positive z-scores observed in the 'lung' clusters confirm that Ahr mRNA is expressed in patient lung tissue. Because the Karlsson pipeline equalizes library depth before z-score transformation, these values permit direct comparison of Ahr abundance to the whole-body baseline captured by the same atlas. Additionally, experimental and clustering details can be found at *Karlsson et al., 2021*.

## In silico analysis of AHR protein abundance in human lung macrophages using HPA immunohistochemistry data

We retrieved the antibody-based tissue microarray summary from *Uhlén et al., 2005*, via the HPA pathology portal, which lists per-tissue immunoreactivity for each antibody as categorical scores ('not detected', 'low', 'medium', 'high'). From the raw TSV file, we report protein expression (*Ahr, Artn, Gfra3, Calca, Vip, Tac1*) across cell types and tissue types. Because the HPA pipeline reports these scores after internal normalization of DAB signal across replicate tissue cores, the categorical value can be compared directly across tissues without further scaling. This processed result confirms constitutive AHR protein presence in lung macrophages in situ. Additionally, experimental and clustering details can be found at *Uhlén et al., 2005*.

## In silico co-expression analysis of AHR and ARTN in human lung macrophages using the CZ CELLxGENE Discover single-cell atlas

We accessed the lung single-cell RNA-sequencing data curated by *Abdulla et al., 2023*, through the CZ CELLxGENE Discover platform and downloaded the cell-level expression matrix together with standardized cell-type annotations. Raw UMI counts were normalized to ln (CPTT+1) values during

ingestion into the atlas, a transformation that equalizes library depth and stabilizes variance while preserving the full gene-cell count matrix for downstream analyses. Additionally, experimental and clustering details can be found at *Abdulla et al., 2023*.

### g:Profiler and GO term

The top 50 DEGs (based on DESeq p-values) from each comparison were submitted to the web-based tool g:Profiler (*Kolberg et al., 2023*) for enrichment analysis (g:GOSt). The resulting pathway enrichments—derived from multiple databases, including Gene Ontologies (GO; covering Molecular Function, Biological Process, and Cellular Component sub-ontologies) (*Ashburner et al., 2000*), the Kyoto Encyclopedia of Genes and Genomes (KEGG) (*Kanehisa et al., 2023*), Reactome (REAC) (*Gillespie et al., 2022*), WikiPathways (WP) (*Martens et al., 2021*), TRANSFAC (TF) (*Wingender, 2008*), miRTarBase (MIRNA), the Comprehensive Resource of Mammalian Protein Complexes (CORUM) (*Tsitsiridis et al., 2023*), and the Human Phenotype Ontology (HP) (*Köhler et al., 2021*)—are listed in *Supplementary file 1*.

### Statistics

p-Values≤0.05 were considered statistically significant. One-way ANOVA and Student's t-tests were conducted using GraphPad Prism, while DESeq2 and Seurat analyses, including their statistical tests, were performed in RStudio.

### Replicates

The number of replicates (n) for each experiment is specified in the figure legends and represents the number of animals for in vivo data. For in vitro experiments, replicates may be culture wells or dishes, animals, FOVs during microscopy, or individual neurons in calcium imaging. All experiments included different preparations from distinct animals to ensure biological reproducibility.

### Exclusion

One replicate in the naïve mice group of the RNA-sequencing analysis (*Figure 2*) was excluded because the principal component analysis (PCA) indicated it as an outlier. No other data were excluded from the study.

## Acknowledgements

ST's work is supported by the Canadian Institutes of Health Research (193741, 407016, 461274, and 461275), the Canadian Foundation for Innovation (44135), the Canadian Cancer Society Emerging Scholar Research Grant (708096), the Knut and Alice Wallenberg Foundation (KAW 2021.0141, KAW 2022.0327), the Swedish Research Council (2022-01661), the Natural Sciences and Engineering Research Council of Canada (RGPIN-2019-06824), and the NIH/NIDCR (R01DE032712). AT's work is supported by the Canadian Institute of Health Research (186176). Salary support for JCW was provided by the Fonds de recherche du Québec – Santé, the Canadian Allergy, Asthma, and Immunology Foundation, Asthma Canada, and CIHR. Salary support for AK was provided by the Canada Graduate Scholarship Masters. IMPB imaging core at McGill University and Nicolas Audet for assistance with IVM experiments.

## Additional information

### Funding

| Funder | Grant reference number | Author |
|---|---|---|
| Canadian Institutes of Health Research | 193741 | Sebastien Talbot |
| Canadian Institutes of Health Research | 407016 | Sebastien Talbot |

| Funder | Grant reference number | Author |
| --- | --- | --- |
| Canadian Institutes of Health Research | 461274 | Sebastien Talbot |
| Canadian Institutes of Health Research | 461275 | Sebastien Talbot |
| Canadian Foundation for Innovation | 44135 | Sebastien Talbot |
| Canadian Cancer Society Emerging Scholar Research Grant | 708096 | Sebastien Talbot |
| Knut and Alice Wallenberg Foundation | KAW 2021.0141 | Sebastien Talbot |
| Knut and Alice Wallenberg Foundation | KAW 2022.0327 | Sebastien Talbot |
| Swedish Research Council | 2022-01661 | Sebastien Talbot |
| Natural Sciences and Engineering Research Council of Canada | RGPIN-2019-06824 | Sebastien Talbot |
| NIH/NIDCR | R01DE032712 | Sebastien Talbot |
| Canadian Institute of Health Research | 186176 | Ajitha Thanabalasuriar |

The funders had no role in study design, data collection and interpretation, or the decision to submit the work for publication.

### Author contributions

Jo-Chiao Wang, Conceptualization, Formal analysis, Investigation, Methodology, Writing – review and editing; Amelia Kulle, Anais Roger, Conceptualization, Formal analysis, Investigation, Methodology; Theo Crosson, Conceptualization, Investigation, Methodology; Amin Reza Nikpoor, Conceptualization, Data curation, Formal analysis, Investigation, Methodology; Surbhi Gupta, Investigation; Moutih Rafei, Conceptualization, Writing – review and editing; Ajitha Thanabalasuriar, Conceptualization, Formal analysis, Funding acquisition, Investigation, Methodology, Writing – review and editing; Sebastien Talbot, Conceptualization, Funding acquisition, Writing – original draft, Writing – review and editing

### Author ORCIDs

Jo-Chiao Wang ⓘ https://orcid.org/0000-0003-0122-4500
Ajitha Thanabalasuriar ⓘ https://orcid.org/0000-0001-6281-3013
Sebastien Talbot ⓘ https://orcid.org/0000-0001-9932-7174

### Ethics

All procedures involving animals adhered to the guidelines of the Canadian Council on Animal Care (CCAC), McGill University (Mgcl-8184) and the Queen's University Animal Care Committee (UACC, protocol 2384).

Reviewer #1 (Public review): https://doi.org/10.7554/eLife.101988.3.sa1
Reviewer #2 (Public review): https://doi.org/10.7554/eLife.101988.3.sa2
Reviewer #3 (Public review): https://doi.org/10.7554/eLife.101988.3.sa3
Author response https://doi.org/10.7554/eLife.101988.3.sa4

# Additional files

### Supplementary files

Supplementary file 1. Differentially expressed genes and pathway analysis of vagal nociceptors in pollution-exacerbated asthma. Naïve 6–10 weeks' male and female *Trpv1*cre::*tdTomato*fl/wt mice

were either subjected to a pollution-exacerbated asthma protocol, to the classic ovalbumin (OVA) protocol, or remained naïve. On day 17, jugular-nodose complex (JNC) neurons were harvested and dissociated, and $Trpv1^+$ (tdTomato$^+$) neurons were FACS-purified to remove stromal and non-peptidergic cells before being processed for RNA-sequencing. The different tabs show the DESeq2 identified for each of these conditions. Other tabs show Gene Ontology (GO) terms enriched in each condition, analyzed using the web-based tool g:Profiler.

MDAR checklist

## Data availability

Bulk RNA-sequencing raw and processed data have been deposited in the NCBI's Gene Expression Omnibus (GSE298583). Processed data can also be accessed in *Supplementary file 1*. Additional information and raw data are available at https://doi.org/10.5061/dryad.hhmgqnkwq.

The following dataset was generated:

| Author(s) | Year | Dataset title | Dataset URL | Database and Identifier |
|---|---|---|---|---|
| Crosson T, Roversi K, Wang J, Talbot S | 2025 | Nociceptor neurons control pollution-mediated neutrophilic asthma; Expression profile of vagal nociceptor neurons in air pollution-exacerbated allergic airway inflammation | https://www.ncbi.nlm.nih.gov/geo/query/acc.cgi?acc=GSE298583 | NCBI Gene Expression Omnibus, GSE298583 |
| Talbot S | 2026 | Nociceptor neurons control pollution-mediated neutrophilic asthma | https://doi.org/10.5061/dryad.hhmgqnkwq | Dryad Digital Repository, 10.5061/dryad.hhmgqnkwq |

The following previously published datasets were used:

| Author(s) | Year | Dataset title | Dataset URL | Database and Identifier |
|---|---|---|---|---|
| Kupari | 2019 | An atlas of vagal sensory neurons and their molecular specialization | https://www.ncbi.nlm.nih.gov/geo/query/acc.cgi?acc=GSE124312 | NCBI Gene Expression Omnibus, GSE124312 |
| Zhao | 2022 | A molecular architecture of the vagal interoceptive system | https://www.ncbi.nlm.nih.gov/geo/query/acc.cgi?acc=GSE192987 | NCBI Gene Expression Omnibus, GSE192987 |
| Zilionis | 2019 | Single cell transcriptomics of human and mouse lung cancers reveals conserved myeloid populations across individuals and species | https://www.ncbi.nlm.nih.gov/geo/query/acc.cgi?acc=GSE127465 | NCBI Gene Expression Omnibus, GSE127465 |

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
