## [Editor Report · eLife Assessment]

This **important** work shows that fine particulate matter exposure to the lungs led to nociceptor-dependent neutrophilic inflammation. Likely macrophage-neuronal crosstalk, via release of artemin from macrophages and activation of Gfra3 on the JNC neuron, potentiated the response. The data **convincingly** strengthens links between pollutants, immune and neural interactions.

---

## [Referee Report · Reviewer #1 (Public review)]

Summary:

In the presented study, the authors aim to explore the role of nociceptors in the fine particulate matter (FPM) mediated Asthma phenotype, using rodent models of allergic airway inflammation. This manuscript builds on previous studies, and identify transciptomic reprogramming and an increased sensitivity of the jugular nodose complex (JNC) neurons, one of the major sensory ganglion for the airways, on exposure to FPM along with Ova during the challenge phase. The authors then use OX-314 a selectively permeable form of lidocaine, and TRPV1 knockouts to demonstrate that nociceptor blocking can reduce airway inflammation in their experimental setup.

The authors further identify the presence of Gfra3 on the JNC neurons, a receptor for the protein Artemin, and demonstrate their sensitivity to Artmein as a ligand. They further show that alveolar macrophages release Artemin on exposure to FPM.

Strengths:

The study builds on results available from multiple previous works, and presents important results which allow insights into the mixed phenotypes of Asthma seen clinically. In addition, by identifying the role of nociceptors, they identify potential therapeutic targets which bear high translational potential.

Weaknesses:

While the results presented in the study are highly relevant, there is a need for further mechanistic dissection to allow better inferences. Currently, certain results seem associative. Also, certain visualisations and experimental protocols presented in the manuscript need careful assessment and interpretation.

While Asthma is a chronic disease, the presented results are particularly important to explore Asthma exacerbations in response to acute exposure to air pollutants. This is relevant in today's age of increasing air pollution and increasing global travel.

Comments on revisions:

Thank you for addressing the suggestions. No further comments.

---

## [Referee Report · Reviewer #2 (Public review)]

Summary:

The authors sought to investigate the role of nociceptor neurons in the pathogenesis of pollution-mediated neutrophilic asthma. The authors overall achieved the aim of demonstrating that nociceptor neurons are important to the pathogenesis of pollution-exacerbated asthma. Their results support their conclusions overall, although there are ways the study findings can be strengthened. This work further evaluates how nociceptor neurons contribute to asthma pathogenesis important for consideration while proposing treatment strategies for under treated asthma endotypes.

Strengths:

The authors utilize TRPV1 ablated mice to confirm the effects of intranasally administered QX-314 utilized to block sodium currents.

Use of intravital microscopy to track alveolar macrophage and neutrophil motility in their model

The authors demonstrate that via artemin, which is upregulated in alveolar macrophages in response to pollution, sensitizes JNC neurons thereby increasing their responsiveness to pollution. Ablation or inactivity of nociceptor neurons prevented the pollution induced increase in inflammation.

Weaknesses:

While neutrophilic, unclear of the endotype of asthma represented by the model

Comments on revisions:

The authors have addressed or commented on all concerns.

---

## [Referee Report · Reviewer #3 (Public review)]

Asthma is a complex disease that includes endogenous epithelial, immune and neural components that respond to environmental stimuli. Small airborne particles with diameters in the range of 2.5 micrometers or less, so-called PM2.5, are thought to contribute to some forms of asthma. These forms of asthma may have neutrophils, eosinophils and macrophages in bronchoalveolar lavage. Here, Wang and colleagues build on a recent model that incorporated PM2.5 which was found to have a neutrophilic component. Wang altered the model to provide an extra kick via the incorporation of ovalbumin. The major strength of this work is that silencing TRPV1-expressing neurons either pharmacologically or genetically, modulated inflammation and the motility of neutrophils. By examining bronchoalveolar lavage fluid, they found not only that levels of a number of cytokines were increased, but also that artemin, a protein that supports neuronal development and function, was elevated, which did not occur in nociceptor- ablated mice. Their data strengthens links between pollutants, immune and neural interactions.

Comments on revisions:

The manuscript has been revised extensively, including the addition of new experiments, such as intravital microscopy. Did the comments from the reviewers, manifest by additional experiments and modifying how some of the data was presented, result in any changes in the hypotheses or the interpretation of such?

---

## [Author Response]

The following is the authors’ response to the original reviews.

**Reviewer #1 (Public review):**
In the presented study, the authors aim to explore the role of nociceptors in the fine particulate matter (FPM) mediated Asthma phenotype, using rodent models of allergic airway inflammation. This manuscript builds on previous studies and identify transcriptomic reprogramming and an increased sensitivity of the jugular nodose complex (JNC) neurons, one of the major sensory ganglia for the airways, on exposure to FPM along with Ova during the challenge phase. The authors then use OX-314 a selectively permeable form of lidocaine, and TRPV1 knockouts to demonstrate that nociceptor blocking can reduce airway inflammation in their experimental setup. The authors further identify the presence of Gfra3 on the JNC neurons, a receptor for the protein Artemin, and demonstrate their sensitivity to Artemin as a ligand. They further show that alveolar macrophages release Artemin on exposure to FPM.

We thank the reviewer for their valuable comments, which have significantly enhanced the quality of our manuscript. A point-by-point rebuttal is provided below.

StrengthThe study builds on results available from multiple previous work and presents important results which allow insights into the mixed phenotypes of Asthma seen clinically. In addition, by identifying the role of nociceptors, they identify potential therapeutic targets which bear high translational potential.WeaknessWhile the results presented in the study are highly relevant, there is a need for further mechanistic dissection to allow better inferences. Currently certain results seem associative. Also, certain visualisations and experimental protocols presented in the manuscript need careful assessment and interpretation. While Asthma is a chronic disease, the presented results are particularly important to explore Asthma exacerbations in response to acute exposure to air pollutants. This is relevant in today's age of increasing air pollution and increasing global travel.MajorThe JNC is a major group of neurons responsible for receiving sensory inputs from the airways. However, the DRG also contains nociceptors and is known to receive afference from the upper airways. An explanation of why the study was restricted to the JNC would be important.

We acknowledge that some afferents to the upper airways do arise from the DRG, specifically in the upper thoracic segments (T1–T5). We have added a statement in the text to note this subset of nociceptive and spinally mediated pathways. However, the preponderance of evidence indicates that the majority of airway and lung afferents (70–80%, sometimes up to 90%) originate from the jugular–nodose complex (JNC). Given this large imbalance—and because our study focuses on the mechanosensory, and chemosensory functions mediated primarily by the JNC—we restricted our analysis to this main vagal pathway. By contrast, DRG innervation, though functionally important for nociception and irritation-related reflexes, accounts for a smaller yet significant (~20–30%) fraction of the total afferent pool. The referenced tracing studies[1,2] support this distribution and are cited to clarify our rationale for emphasizing the JNC in our work.

Similarly, the role of the Artemin in the study remains associative. The study results present that Artemin sensitize nociceptors to lead to an increased inflammatory response (Supplementary Figure 2), however, both upstream and downstream evidence for this inference needs to be dissected further. For instance, the evidence for the role of Artemin in the model comes from ex vivo experiments with alveolar macrophages, but not in the experimental model created. Blocking or activation experiments could be performed, along with investigating the change in the total number of nociceptors with Artemin exposure. Similarly, the downstream effects of the potential Artemin-mediated JNC stimulation should be explored in the context of this experimental setup. A detailed dissection of the mechanisms is important. Additionally, it is also important to discuss the hypothesis leading to the selection of Artemin as a target, which currently seems arbitrary.

Our data show that exogenous (i) OVA-FPM exposed AM secrete Artemin and that (ii) recombinant Artemin can sensitize nociceptors, potentially heightening the inflammatory response. As suggested, we agree that more upstream and downstream evidence is needed for definitive mechanistic insight. In response, we have expanded our experiments to include intravital microscopy, which demonstrates impaired motility of alveolar macrophages and neutrophils in nociceptor-ablated mice, suggesting a bidirectional crosstalk between AMs and nociceptor neurons.

In future studies, we will perform blocking or activation studies to clarify Artemin’s in vivo effects and confirm its role in modulating airway nociceptors. We also recognize the importance of examining whether Artemin exposure alters the phenotype of these neurons and lung innervation density. As recommended, we plan targeted interventions (e.g., Artemin-neutralizing antibodies or overexpression strategies) to delineate the mechanisms by which Artemin-mediated nociceptor stimulation influences the local inflammatory environment.

We have expanded our discussion to clarify that Artemin is a recognized growth factor known to sensitize certain sensory neurons, including those responsive to tissue injury and inflammation. This literature-based rationale guided our hypothesis that Artemin might increase nociceptor reactivity in the lung and thereby influence alveolar macrophage function. By combining ex vivo and intravital approaches, we have begun to map these interactions but agree that further in vivo studies are necessary to confirm causality, dissect signal transduction pathways, and fully validate Artemin’s contributions to AM–nociceptor crosstalk. We have revised our manuscript accordingly to highlight these limitations.

A deeper exploration of the inflammatory parameters could be performed. The multiplex analysis of the cytokine analysis shows a reduction in certain cytokines like IL-6 and MCP (figure 3F), which needs to be discussed. Additionally, investigating the change in proportions of the different immune cell populations is important, which currently restricts the eosinophil and neutrophil counts in the BAL. This is also important as the study builds on work from Prof. Chang's group, which also identified the expansion of an invariant iNKT cell population by FPM, regulatory in nature. Adding data on airway hyperresponsiveness, if possible, would be a welcome addition, considering Asthma as the disease context.

We thank the reviewer for highlighting the need for a more comprehensive exploration of inflammatory parameters. To address these concerns:

(1) Cytokine Analysis: We re-ran all statistical analyses, including the CBA and ELISA assays, and confirmed that TNFα and Artemin are the only differentially expressed cytokines across experimental groups. We have expanded the Discussion to emphasize TNFα’s role in this context.

(2) Immune Cell Profiling in BALF: Our data show that co-exposure with FPM exacerbates CD45+ cells, eosinophil, neutrophil, T-cells and monocyte infiltration. Notably, CD45+ cells and neutrophils were the only population reduced under nociceptor neuron loss-of-function conditions (QX314–treated or TRPV1-DTA mice, Author response image 1).

Of note, we also confirmed these data using intravital imaging and in a second line of nociceptor ablated mice (NaV1.8DTA). We are aware of Prof. Chang’s work suggesting expansion of an invariant iNKT cell population this population in future

(3) Airway Hyperresponsiveness (AHR): We recognize that adding AHR data would strengthen the asthma-related context. Unfortunately, we are not currently equipped to perform AHR measurements, but we intend to include this in future experiments to provide a more complete assessment of airway function.

**Author response image 1. sa4fig1:** 

The authors could revisit the data presented in terms of visualization. For instance, the pooled data presented in some of the figures is probably leading to a wide variation which makes interpretation more difficult. Presenting data separately for each experimental replicate might help the reader. This is also important considering the possible variation seen between experiments (for instance, in Figure 3A and 3C and 3B and 3D, the neutrophil and eosinophil panels for the same groups seem to have an almost 2-fold difference.). Similarly, in the cytokine analysis, the authors have used a common axis for depicting all cytokine values which leads to difficulties in interpretation (Figure 3F). Analysis of the RNA seq results and the DEGs could be revisited to include pathway analysis etc (Figure 2), and the supplementary information could include detailed lists of the major target genes.

To address this query, we have completely reformatted all graphs and included both gene lists and lists of enriched pathways for all three comparisons in Supplementary Table 1. We also confirmed our flow cytometry analysis functionally by performing intravital imaging.

The authors should also consider citing the previous experimental setup used for some particular protocols. For instance, the use of the specified protocol for OVA in a C57 background needs to be justified, as there are various protocols reported in the literature. Additionally, doses used in some experiments seem arbitrary (The FPM and Artemin exposure in Figure 4). Depicting the dose-response curve or citing previous literature for the same would be important. Similarly, different sample sizes seen in experiments should be explained, whether they are due to mortality, failure to exhibit phenotypes, or due to technical failures. The RNA seq experiment mentions only 2 biological replicates in one of the groups which should be addressed either by increasing the sample size or by replicating the experiment. Moreover, nested comparisons in experiments performed for Figure 1 need to be performed. Neurons isolated from each mouse should be maintained and analysed separately to retain biological replicates to better represent the heterogeneity.

We appreciate the request for clarity regarding the experimental protocols and sample sizes:

OVA Model in C57BL/6 Mice: We adapted a previously published OVA protocol in C57BL/6 mice[3-5] (PMID: 39661516), which uses two doses of sensitization to compensate for the lower Th2 response compared to BALB/c[6]. We increased the dose of OVA (100 µg) because our initial experiments produced low eosinophil infiltration. Although this dosage is on the higher side, some studies have noted local IFNγ induction in C57BL/6 mice; however, we did not detect IFNγ in our setup.

FPM and Artemin Doses: We did not perform a full dose-response assay for FPM and Artemin but used 100 ng/mL as reported in prior literature, where TRPA1 and TRPV1 mRNA were upregulated after 18 hours of incubation[7]. This reference has been added for clarity.

Sample Sizes and Exclusions: One control mouse was excluded from the RNA-seq experiment because a parallel PCA analysis indicated it was an outlier. This was the only exclusion in the study, and this have been indicated in the method section of the article.

Nested Comparisons and Biological Replicates: We reanalyzed the relevant data with a nested one-way ANOVA and updated the figures accordingly. Neurons isolated from each mouse were first averaged to preserve biological replicates and capture potential heterogeneity; and data was analysed on the per mouse averages.

The manuscript should be more detailed regarding the statistics employed. Currently, there is a section mentioned in the methods section, but details of corrections employed and specific stats for specific experiments should be described. There are also some minor grammatical errors and incomplete sentences in the manuscript which should be corrected. The authors should also consider a more expansive literature review in the introduction/discussion sections.

We have updated the figure legends and methods to include more detailed information on the specific statistical tests used for each experiment. In addition, we have fixed minor grammatical errors and incomplete sentences throughout the manuscript. Finally, we have expanded our Introduction and Discussion to include additional references and a broader literature context.

**Reviewer #2 (Public review):**
The authors sought to investigate the role of nociceptor neurons in the pathogenesis of pollutionmediated neutrophilic asthma.

We thank the reviewer for their valuable comments, which have significantly enhanced the quality of our manuscript. A point-by-point rebuttal is provided below.

StrengthThe authors utilize TRPV1 ablated mice to confirm effects of intranasally administered QX-314 utilized to block sodium currents. The authors demonstrate that via artemin, which is upregulated in alveolar macrophages in response to pollution, sensitizes JNC neurons thereby increasing their responsiveness to pollution. Ablation or inactivity of nociceptor neurons prevented the pollution induced increase in inflammation.WeaknessWhile neutrophilic, the model used doesn't appear to truly recapitulate a Th2/Th17 phenotype. No IL-17A is visible/evident in the BALF fluid within the model. (Figure 3F). Unclear of the relevance of the RNAseq dataset, none of the identified DEGs were evaluated in the context of mechanism. The authors overall achieved the aim of demonstrating that nociceptor neurons are important to the pathogenesis of pollutionexacerbated asthma. Their results support their conclusions overall, although there are ways the study findings can be strengthened. This work further evaluates how nociceptor neurons contribute to asthma pathogenesis important for consideration while proposing treatment strategies for undertreated asthma endotypes.MajorUtilizing a different model, one using house dust mite or alternaria alternata or similar that is able to induce a true Th2/th17 type response that is also more translatable to humans for confirmation.

We appreciate the suggestion to use additional allergen models. In a pilot study, we did observe increased Artemin in the BALF of house dust mite–treated mice, although the levels were low under our current dosing schedule (20 µg/dose daily from Day 0–4 and Day 7–9, with sacrifice on Day 10; Auhtor response image 2). Conversely, using an Alternaria alternata model at 100 µg/dose daily from Day 0–2 (sacrificed on Day 3) did not yield a detectable increase in Artemin. We suspect these findings may reflect the specific dose and timing used. We plan to refine our protocols (e.g., longer exposures or higher doses) for HDM and/or Alternaria to better model a Th2/Th17 response and further validate our observations in a setting more translatable to human asthma.

**Author response image 2. sa4fig2:** 

Additional analysis, maybe pathway analysis on the RNAseq dataset presented in Figure 2. Unclear how these genes are relevant/how they affect functionality. At present it is acceptable to say they are transcriptionally reprogramed, but no protein evaluation is provided which would get more at function, however, the authors do show some functional data in Figure 1, so maybe this could somehow be discussed/related to Figure 2.

We have expanded our RNA-seq analysis to include gene lists and enriched pathways for all three comparisons in Supplementary Table 1. We have also revised our discussion to align these transcriptomic changes with the functional data shown in Figure 1. While we have not yet performed protein-level validation for all identified genes, the patterns observed in our RNA-seq dataset suggest pathways potentially tied to nociceptor activation and the downstream inflammatory response. We plan to conduct targeted protein analyses in future studies to further substantiate these findings.

Histology and localization of neutrophils/nociceptor neurons/alveolar macrophages would enhance the study findings.

We appreciate the reviewer’s suggestion to include histological data showing the distribution of neutrophils, nociceptor neurons, and alveolar macrophages. While we have not yet performed detailed histological staining of these cell types, we have added live in-vivo intravital microscopy data (Figure 4) that illustrate impaired AM and neutrophil motility in nociceptor-ablated mice. We plan to include additional histological analyses in future studies to further localize these cells in the lung tissue.

Minor:The first 3 figures are small and hard to read.

We have enlarged Figures 1 and 3 in the revised manuscript to improve readability. We have also added the corresponding gene lists and enriched pathways to Supplementary Table 1 for clarity.

The figures are mislabeled in the text. Figure 2 is discussed twice in two different contexts; the second mention is supposed to be labeled as Figure 2.

We corrected the mislabeled figures in the text, ensuring that each figure is referenced accurately.

Figure 4 isn't cited in the text. I think it is supposed to be referenced in the paragraph before the discussion starts and is currently labeled as Figure 1.

We have updated the text to properly cite Figure 4 in the relevant paragraph before the Discussion begins, rather than labeling it as Figure 1.

Notating which statistical analysis was used with each figure/subfigure would be beneficial. Also, it's important to notate if the data was analyzed for multiple comparisons.

We have revised each figure/subfigure legend to specify the statistical tests used, including information on whether corrections for multiple comparisons were applied. This provides a clearer understanding of how each dataset was analyzed.

**Reviewer #3 (Public review):**
Asthma is a complex disease that includes endogenous epithelial, immune, and neural components that respond awkwardly to environmental stimuli. Small airborne particles with diameters in the range of 2.5 micrometers or less, so-called PM2.5, are generally thought to contribute to some forms of asthma. These forms of asthma may have increased numbers of neutrophils and/or eosinophils present in bronchoalveolar lavage fluid and are difficult to treat effectively as they tend to be poorly responsive to steroids. Here, Wang and colleagues build on a recent model that incorporated PM2.5 which was found to have a neutrophilic component. Wang altered the model to provide an extra kick via the incorporation of ovalbumin. Building on their prior expertise linking nociceptors and inflammation, they find that silencing TRPV1-expressing neurons either pharmacologically or genetically, abrogated inflammation and the accumulation of neutrophils. By examining bronchoalveolar lavage fluid, they found not only that levels of the number of cytokines were increased, but also that artemin, a protein that supports neuronal development and function, was elevated, which did not occur in nociceptor-ablated mice. They also found that alveolar macrophages exposed to PM2.5 particles had increased artemin transcription, suggesting a further link between pollutants, and immune and neural interactions.

We thank the reviewer for their valuable comments, which have significantly enhanced the quality of our manuscript. A point-by-point rebuttal is provided below.

WeaknessThere are substantial caveats that must be attached to the suggestions by the authors that targeting nociceptors might provide an approach to the treatment of neutrophilic airway inflammation in pollutiondriven asthma in general and wildfire-associated respiratory problems in particular.These caveats include the uncertainty of the relevance of the conventional source of PM2.5, to pollution and asthma. According to the National Institute of Standards and Technology (NIST), the standard reference material (SRM) 2786 is a mix obtained from an air intake system in the Czech Republic. It is not clear exactly what is in the mix, and a recent bioRxiv preprintpreprint, reveals the presence of endotoxin. Care should thus be taken in interpreting data using particulate matter. Regarding wildfires, there is data that indicates that such exposure is toxic to macrophages. What impact might that then have on the production of cytokines, and artemin, in humans?

We recognize the potential limitations of using SRM2786 (obtained from a Czech air-intake system) as a model for realworld PM2.5 exposure. Our rationale for choosing SRM2786 is that it is commercially available and represents a broad spectrum of ambient air pollutants, in contrast to more specialized sources like diesel exhaust particles. However, we acknowledge in the discussion the presence of endotoxin in SRM2786, as suggested by recent reports, and agree that this may influence immune responses and should be considered when interpreting our data.

Regarding wildfire-associated exposure, we are aware that certain components of wildfire smoke can be toxic to macrophages. We do not think this play a significant role in the current study design as number of AMs, as determined by flow cytometry and intravital microscopy, are similar when comparing OVA-exposed mice to OVA-FPM exposed animals. Thus, these results rule out significant AM toxicity by FPM.

Ultimately, while our findings suggest that modulating nociceptor activity may reduce neutrophilic inflammation, we emphasize that additional research—including different PM2.5 sources, validation of endotoxin levels, and in vivo confirmation in human-relevant models—is necessary before drawing definitive conclusions about treating pollutiondriven asthma or wildfire-induced respiratory problems.

The Introductory paragraph implies links between wildfire events, particular exposure, and neutrophilic asthma. I am not aware of such a link having been established, in which case the paragraph needs revision. In the paragraph that begins with 'Urban pollution', it is suggested that eosinophilic asthma is treatment responsive in comparison to the neutrophilic form. That may not be the case, and they may often these cellular components may occur together. In much of the manuscript, there is a mismatch between the text and the figure numbers. For example, in the Results, Figure 2 should be Figure 3 some of the time, and Figure 3 is actually Figure 4, while the reference to Figure 1F-H is Figure 4H. Please check carefully.

(a) Introduction Paragraph and Wildfire–Neutrophilic Asthma Link

We add references to the introduction to support the link between wildfire, respiratory symptoms and the link to neutrophilic asthma [8-12].

(b) Distinction Between Eosinophilic and Neutrophilic Asthma

We recognize that eosinophilic and neutrophilic airway infiltrates can co-occur in the same individual and that treatment responsiveness can vary considerably. Our intention was to note that conventional asthma therapies (e.g., inhaled corticosteroids) are generally more effective for eosinophilic-driven disease than for neutrophilic phenotypes, but we agree that these inflammatory endotypes often overlap in clinical practice. We have revised the text in the “Urban pollution” section to acknowledge this complexity and to clarify that inflammatory cell populations in asthma are not always discrete.

Figure Numbering and Text–Figure Mismatch

We sincerely apologize for the confusion caused by mismatched figure labels and references in the Results section. We have carefully reviewed and corrected all figure references throughout the manuscript to ensure accuracy.

References

(1) Kim, S. H. et al. Mapping of the Sensory Innervation of the Mouse Lung by Specific Vagal and Dorsal Root Ganglion Neuronal Subsets. eNeuro 9 (2022). https://doi.org/10.1523/ENEURO.0026-22.2022

(2) McGovern, A. E. et al. Evidence for multiple sensory circuits in the brain arising from the respiratory system: an anterograde viral tract tracing study in rodents. Brain Struct Funct 220, 3683-3699 (2015). https://doi.org/10.1007/s00429-014-0883-9

(3) Shen, C.-C., Wang, C.-C., Liao, M.-H. & Jan, T.-R. A single exposure to iron oxide nanoparticles attenuates antigen-specific antibody production and T-cell reactivity in ovalbumin-sensitized BALB/c mice. International journal of nanomedicine, 1229-1235 (2011).

(4) Delayre-Orthez, C., De Blay, F., Frossard, N. & Pons, F. Dose-dependent effects of endotoxins on allergen sensitization and challenge in the mouse. Clinical & Experimental Allergy 34, 1789-1795 (2004).

(5) Morokata, T., Ishikawa, J. & Yamada, T. Antigen dose defines T helper 1 and T helper 2 responses in the lungs of C57BL/6 and BALB/c mice independently of splenic responses. Immunology letters 72, 119-126 (2000).

(6) Li, L., Hua, L., He, Y. & Bao, Y. Differential effects of formaldehyde exposure on airway inflammation and bronchial hyperresponsiveness in BALB/c and C57BL/6 mice. PLoS One 12, e0179231 (2017).

(7) Ikeda-Miyagawa, Y. et al. Peripherally increased artemin is a key regulator of TRPA1/V1 expression in primary afferent neurons. Molecular pain 11, s12990-12015-10004-12997 (2015).

(8) Baan, E. J. et al. Characterization of Asthma by Age of Onset: A Multi-Database Cohort Study. J Allergy Clin Immunol Pract 10, 1825-1834 e1828 (2022). https://doi.org/10.1016/j.jaip.2022.03.019

(9) de Nijs, S. B., Venekamp, L. N. & Bel, E. H. Adult-onset asthma: is it really different? Eur Respir Rev 22, 44-52 (2013). https://doi.org/10.1183/09059180.00007112

(10) Gianniou, N. et al. Acute effects of smoke exposure on airway and systemic inflammation in forest firefighters. J Asthma Allergy 11, 81-88 (2018). https://doi.org/10.2147/JAA.S136417

(11) Noah, T. L., Worden, C. P., Rebuli, M. E. & Jaspers, I. The Effects of Wildfire Smoke on Asthma and Allergy. Curr Allergy Asthma Rep 23, 375-387 (2023). https://doi.org/10.1007/s11882-023-01090-1

(12) Wilgus, M. L. & Merchant, M. Clearing the Air: Understanding the Impact of Wildfire Smoke on Asthma and COPD. Healthcare (Basel) 12 (2024). https://doi.org/10.3390/healthcare12030307